# Accelerating functional gene discovery in osteoarthritis

Natalie C. Butterfield [1], Katherine F. Curry[1], Julia Steinberg [2,3,4], Hannah Dewhurst[1], Davide Komla-Ebri [1], Naila S. Mannan[1], Anne-Tounsia Adoum[1], Victoria D. Leitch [1], John G. Logan[1], Julian A. Waung[1], Elena Ghirardello[1], Lorraine Southam[2,3], Scott E. Youlten [5], J. Mark Wilkinson [6,7], Elizabeth A. McAninch [8], Valerie E. Vancollie [3], Fiona Kussy[3], Jacqueline K. White[3,9], Christopher J. Lelliott [3], David J. Adams [3], Richard Jacques [10], Antonio C. Bianco[11], Alan Boyde [12], Eleftheria Zeggini [2,3], Peter I. Croucher [5], Graham R. Williams [1,13✉] & J. H. Duncan Bassett [1,13✉]

Osteoarthritis causes debilitating pain and disability, resulting in a considerable socio-economic burden, yet no drugs are available that prevent disease onset or progression. Here, we develop, validate and use rapid-throughput imaging techniques to identify abnormal joint phenotypes in randomly selected mutant mice generated by the International Knockout Mouse Consortium. We identify 14 genes with functional involvement in osteoarthritis pathogenesis, including the homeobox gene *Pitx1*, and functionally characterize 6 candidate human osteoarthritis genes in mouse models. We demonstrate sensitivity of the methods by identifying age-related degenerative joint damage in wild-type mice. Finally, we phenotype previously generated mutant mice with an osteoarthritis-associated polymorphism in the *Dio2* gene by *CRISPR/Cas9* genome editing and demonstrate a protective role in disease onset with public health implications. We hope this expanding resource of mutant mice will accelerate functional gene discovery in osteoarthritis and offer drug discovery opportunities for this common, incapacitating chronic disease.

[1] Molecular Endocrinology Laboratory, Department of Metabolism, Digestion and Reproduction, Imperial College London, London W12 0NN, UK. [2] Institute of Translational Genomics, Helmholtz Zentrum München – German Research Center for Environmental Health, 85764 Neuherberg, Germany. [3] Wellcome Trust Sanger Institute, Hinxton, Cambridge CB10 1SA, UK. [4] Cancer Council NSW, Sydney, NSW 2000, Australia. [5] The Garvan Institute of Medical Research and St. Vincent's Clinical School, University of New South Wales Medicine, Sydney, NSW 2010, Australia. [6] Department of Oncology and Metabolism, University of Sheffield, Sheffield S10 2RX, UK. [7] Centre for Integrated Research into Musculoskeletal Ageing and Sheffield Healthy Lifespan Institute, University of Sheffield, Sheffield S10 2TN, UK. [8] Division of Endocrinology and Metabolism, Rush University Medical Center, Chicago, IL 60612, USA. [9] The Jackson Laboratory, Bar Harbor, ME 04609, USA. [10] School of Health and Related Research (ScHARR), University of Sheffield, Sheffield S1 4DA, UK. [11] Section of Adult and Pediatric Endocrinology, Diabetes & Metabolism, Department of Medicine, University of Chicago, Chicago, IL 60637, USA. [12] Dental Physical Sciences, Queen Mary University of London, Mile End Road, London E1 4NS, UK. [13] These authors contributed equally: Graham R. Williams, J. H. Duncan Bassett. ✉email: graham.williams@imperial.ac.uk; d.bassett@imperial.ac.uk

Osteoarthritis is the commonest cause of joint destruction, pain, and disability. Joint replacement for end-stage disease remains the only treatment, leading to an escalating healthcare crisis in our obese and ageing society. Osteoarthritis is a complex trait and the 86 reported genome-wide associated loci explain only a small proportion of its heritability, which is estimated between 40 and 70%[1–3].

Osteoarthritis is characterized by articular cartilage damage and loss, together with structural abnormalities of subchondral bone and low-grade chronic joint inflammation. It is unknown which of these processes trigger disease or which represent secondary responses to joint destruction[4]. It is also uncertain whether the pathogenesis of osteoarthritis reflects an abnormal response to injury involving defective stem cell recruitment and abnormal cell proliferation, differentiation, metabolism, apoptosis, and senescence[5,6].

Chondrocytes in healthy articular cartilage are resistant to terminal differentiation, whereas they revert to a developmental program following injury, in which they proliferate and undergo hypertrophic differentiation with accelerated cartilage mineralization[7]. Osteoarthritis pathogenesis involves cross-talk between the synovium, articular cartilage, and subchondral bone[8], although the timing of bone remodeling relative to cartilage degradation remains uncertain[9]. Nevertheless, increases in apoptotic and senescent chondrocytes are triggered by processes including endoplasmic reticulum stress[10]. Senescent cells express a secretory phenotype that contributes to inflammation, vascular invasion[11], and cartilage breakdown via key pathways that stimulate matrix metalloproteases[12] and aggrecan-specific proteinases[13].

Despite the profound clinical and socioeconomic impacts of osteoarthritis, our understanding of its genetic basis is in its infancy. We hypothesize that accelerating gene discovery in osteoarthritis will increase understanding of joint physiology and disease pathogenesis and facilitate identification of drug targets that prevent or delay joint destruction. Studies of extreme phenotypes in humans have underpinned identification of the molecular basis of single gene disorders and mechanisms of complex disease and resulted in new treatments[14,15]. Analogous to our gene discovery studies in osteoporosis[16–20], we propose that a joint-specific extreme phenotype screen in mutant mice will accelerate functional gene discovery in osteoarthritis.

Mutant mice are generated at the Sanger Institute as part of the International Mouse Phenotyping Consortium (IMPC). Mice undergo broad phenotyping using the International Mouse Phenotyping Resource of Standardized Screen (IMPReSS) that is completed at 16 weeks of age when tissues are harvested for further analysis. In the Origins of Bone and Cartilage Disease (OBCD) Project we collaborate with IMPC and receive knee joints for analysis. Rapid-throughput phenotyping of the mouse knee requires quantitative imaging; this presents a complex and unsolved challenge that relates to anatomical size, three-dimensional complexity, image resolution, and the necessity to maintain joints in their native fully hydrated state.

Here we present the invention, optimization, validation, and application of a rapid-throughput multimodality imaging pipeline to phenotype the mouse knee. We analyze 50 randomly selected mouse lines, identifying seven (14%) with markedly abnormal phenotypes. A systematic prioritization strategy identifies seven further lines, resulting in 14 genes (28%) with evidence for a functional role in osteoarthritis pathogenesis. The four leading candidates are *Pitx1*, *Bhlhe40*, *Sh3bp4,* and *Unk*. We interrogate the database of joint phenotypes from randomly selected mouse lines with 409 genes differentially expressed in human osteoarthritis cartilage. This results in an enriched yield of abnormal joint phenotypes in six (75%) of eight lines for which data are

available, including *Unk*. We then apply the pipeline to characterize the early features of age-related joint degeneration in 1-year old mice and demonstrate its sensitivity to detect disease onset as well as surgically provoked late-stage disease, paving the way for application to analysis of drug intervention studies. Finally, we phenotype previously generated *CRISPR/Cas9* mutant mice with a Thr92Ala polymorphism in the *Dio2* gene that is orthologous to the human variant associated with osteoarthritis susceptibility. The Ala92 allele confers protection against early-onset osteoarthritis, challenging current understanding with implications for public health.

## Results

**Invention and optimization of imaging methods**. We established a rapid-throughput joint phenotyping pipeline (OBCD joint pipeline), which applies three complementary imaging approaches to characterize features of osteoarthritis in mutant mice that include articular cartilage damage and loss, together with abnormalities of subchondral bone structure and mineralization (Fig. 1). Iodine contrast-enhanced micro-computerized tomography (ICEµCT) was developed to determine articular cartilage volume (Cg.V), median thickness (Median Cg.Th), maximum thickness (Max Cg.Th), subchondral bone volume/tissue volume (SC BV/TV), trabecular thickness (SC Tb.Th), trabecular number (SC Tb.N), and tissue mineral density (SC TMD). Joint surface replication (JSR) was invented and optimized to quantify articular cartilage surface damage. Subchondral bone X-ray microradiography (scXRM) was developed from previous protocols[21,22] to determine subchondral bone mineral content (SC BMC) (Figs. 1, 2 and Supplementary Fig. 1).

**Validation of imaging methods**. The OBCD joint pipeline methods were validated by comparison with Osteoarthritis Research Society International (OARSI) histological scoring[23] of knees from 22-week-old wild-type (WT) mice, 12 weeks after destabilization of the medial meniscus (DMM)[13,24]. One cohort of WT mice ($n = 16$) was phenotyped using ICEµCT, JSR, and scXRM in the OBCD joint pipeline, and a second cohort ($n = 11$) was analyzed by OARSI scoring (Fig. 1). DMM surgery resulted in decreased Cg.V and Max Cg.Th with a marked increase in Cg. damage (Fig. 3). This cartilage destruction was accompanied by increased SC BV/TV, increased SC Tb.Th, decreased SC Tb.N and increased SC BMC (Fig. 3 and Supplementary Fig. 2). These abnormalities were consistent with extensive cartilage damage, synovitis, and osteophyte formation demonstrated by OARSI-scored histology together with ICEµCT and JSR (Fig. 3 and Supplementary Fig. 2, Supplementary Fig. 3 and Supplementary Data 1, Supplementary Data 2).

OARSI analysis was also undertaken on three joints that were phenotyped in the OBCD joint pipeline and found to have features indicating mild, intermediate and severe osteoarthritis following DMM surgery. OARSI scoring was concordant with the severity of abnormalities identified by ICEµCT, JSR, and scXRM, thus validating the use of these imaging modalities to characterize osteoarthritis (Supplementary Fig. 4).

**Definition of WT reference ranges**. Phenotype data sets for all parameters were obtained from 100 16-week-old male WT mice. Reference ranges, coefficients of variation, estimates of skewness and kurtosis, normality, repeatability, and power calculations were determined (Supplementary Data 3). The reference range for each parameter was defined as either (i) the mean ±2.0 standard deviations for normally distributed data, or (ii) the median and 2.5–97.5th percentile range for non-normally distributed data (Supplementary Fig. 1).

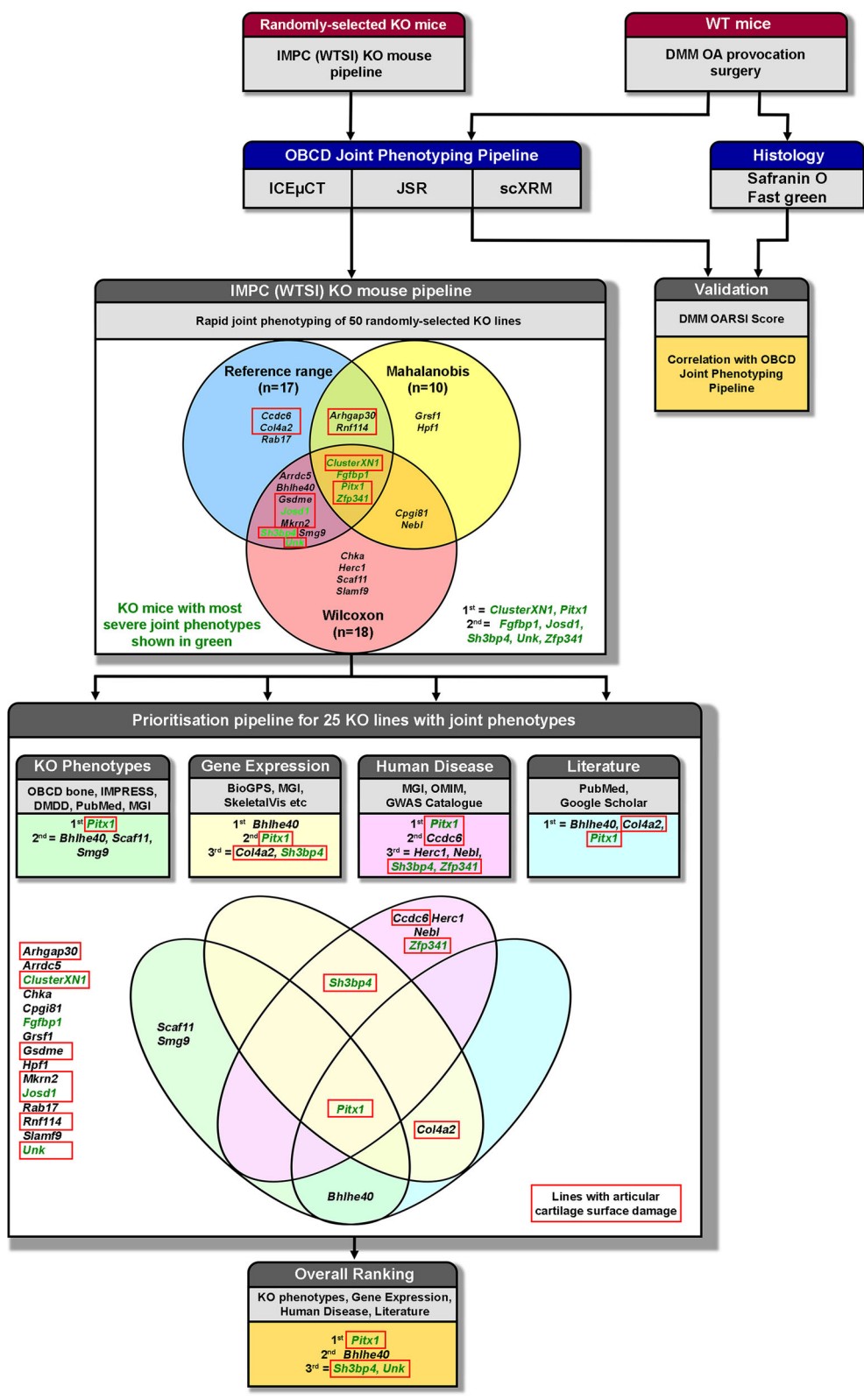

**Identification of osteoarthritis genes**. To identify genes that cause osteoarthritis, we used the OBCD joint pipeline to analyze knees from 16-week-old male mice ($n = 3-7$) from 50 randomly selected mutant lines generated in an identical *C57BL6/N; C57BL6/NTac* genetic background (Supplementary Data 4).

Rigorous statistical approaches were used to determine which lines displayed robust outlier phenotypes (Fig. 1, Supplementary

Data 4 and 5). Lines with outlier phenotypes were identified: (i) when the mean value for an individual parameter was outside the WT reference range; (ii) by using the Wilcoxon rank sum statistical test to analyze all individual phenotype measurements rather than only mean values, with a Bonferroni multiple-testing correction for the effective number of tests (Supplementary Data 5); and (iii) by calculation of Mahalanobis distances to

**Fig. 1 Rapid-throughput joint phenotyping identifies osteoarthritis genes.** Knee joints from randomly selected 16-week-old mutant mice generated at the Wellcome Trust Sanger Institute (WTSI) for the International Mouse Phenotyping Consortium (IMPC) are analyzed by three imaging modalities; iodine contrast-enhanced micro-computerized tomography (ICEμCT), joint surface replication (JSR) and subchondral bone X-ray microradiography (scXRM). The methods were validated by comparison with scoring of histological sections (Osteoarthritis Research Society International (OARSI) histology after destabilization of the medial meniscus (DMM) provocation surgery). Twenty-five mouse lines (three-way Venn diagram) had outlier phenotypes relative to wild-type reference data following statistical analyses. Further prioritization was based on additional skeletal abnormalities identified in mutant mice, gene expression, association with human disease and literature searching. Boxes and four-way Venn diagram show top-ranked genes in each category. *Pitx1*, *Bhlhe40*, *Sh3bp4*, and *Unk* were the top-ranked genes. Green text: most severely abnormal joint phenotypes. Red boxes indicate mutant lines with increased articular cartilage surface damage. Source data are provided as a Source Data file.

ensure that lines with significantly abnormal phenotypes were not overlooked when they resulted from simultaneous but smaller variances in multiple phenotype parameters. Seventeen lines (34%) had at least one abnormal phenotype parameter outside the reference range, 12 (24%) of which had increased cartilage surface damage. Eighteen lines (36%) were outliers after statistical analysis using the Wilcoxon test with Bonferroni correction, and 10 (20%) were outliers following Mahalanobis analysis. Overall, 25 individual lines (50%) had an abnormal joint phenotype based on these criteria (Fig. 1). Increased body weight was not present in any mutant line (Supplementary Data 4). Furthermore, body weight did not correlate with any joint phenotype parameter in WT mice ($P < 0.05$, Spearman correlation, Supplementary Data 6).

The effect of gene deletion on phenotype severity was assessed by scoring whether joint abnormalities were (i) reference range outliers, (ii) outliers based on the Bonferroni-corrected Wilcoxon test, (iii) outliers after Mahalanobis analysis, and by considering whether joint pathology included abnormalities of cartilage morphology, cartilage integrity and/or subchondral bone structure (Supplementary Data 7). Of the 25 lines with an outlier phenotype, the top seven ranked lines with the most severe abnormalities were a cluster of six microRNAs (*ClusterXN1: miR-106a, miR-18b, miR-19b-2, miR-20b, miR-92-2, miR-363*), paired like homeodomain 1 (*Pitx1*), fibroblast growth factor binding protein 1 (*Fgfbp1*), Josephin domain containing 1 (*Josd1*), SH3 domain binding protein 4 (*Sh3bp4*), unkempt family zinc finger (*Unk*), and zinc finger protein 341 (*Zfp341*) (Fig. 1, Supplementary Data 7).

**Prioritization of 25 mouse lines with outlier phenotypes.** An informatics strategy was used to investigate biological plausibility and prioritize the 25 candidate genes based on (i) additional skeletal consequences of gene deletion, (ii) gene expression in skeletal cells and tissues, (iii) association with human disease, and (iv) structured literature searching. Each criterion was assigned a score and scores were summed to rank genes in order of priority.

Additional skeletal consequences of gene deletion were investigated by: (i) identifying whether mutant mice had abnormalities of bone structure and strength[16], (ii) IMPC IMPReSS phenotype screening, (iii) analysis of the skeleton in embryos from lines in which homozygous gene deletion was lethal or resulted in sub-viability[25], and (iv) determining whether mutations of the same gene had skeletal abnormalities identified in Mouse Genome Informatics (MGI) databases or the published literature. The top-ranked genes with major effects on skeletal phenotype were *Pitx1*, basic helix-loop-helix family member e40 (*Bhlhe40*), SR-related CTD associated factor 11 (*Scaf11*), and SMG9 nonsense-mediated mRNA decay factor (*Smg9*) (Fig. 1, Supplementary Data 7).

Gene expression in skeletal cells and tissues was investigated by interrogation of (i) MGI and BioGPS[26] databases, and (ii) transcriptome datasets from human chondrocytes and

cartilage[27,28], mouse and human osteoblasts[26,28], mouse osteocytes[19], and mouse and human osteoclasts[26,28]. The top-ranked genes with expression enriched in cartilage and bone compared with non-skeletal tissues were *Bhlhe40*, *Pitx1*, collagen type IV alpha 2 chain (*Col4a2*) and *Sh3bp4* (Fig. 1, Supplementary Data 7).

Association with human disease was investigated by searching MGI and Online Mendelian Inheritance in Man (OMIM) databases to determine whether mutations in any of the 25 genes resulted in monogenic diseases affecting the skeleton. To determine whether any gene loci were associated with arthritis or other skeletal phenotypes, we interrogated the European Bioinformatics Institute GWAS catalog. The top-ranked genes associated with arthritis and skeletal disease in humans were *Pitx1*, coiled-coil domain containing 6 (*Ccdc6*), HECT and RLD domain containing E3 ubiquitin protein ligase family member 1 (*Herc1*), nebulette (*Nebl*), *Sh3bp4*, and *Zfp341* (Fig. 1, Supplementary Data 7).

Structured literature searching of PubMed and Google Scholar databases identified *Bhlhe40*, *Col4a2* and *Pitx1* as the top-ranked candidate genes with publications related to arthritis or skeletal cell biology (Fig. 1, Supplementary Data 7).

Together, consideration of additional skeletal phenotypes, gene expression, association with human disease and the published literature prioritized 10 lines (*Pitx1*, *Bhlhe40*, *Scaf11*, *Smg9*, *Col4a2*, *Sh3bp4*, *Ccdc6*, *Herc1*, *Nebl* and *Zfp341*). Eleven (44%) of the 25 randomly selected candidates with outlier phenotypes (*Arhgap30*, *Arrdc5*, *ClusterXN1*, *Cpgi81*, *Gsdme*, *Hpf1*, *Josd1*, *Mkrn2*, *Scaf11*, *Slamf9*, *Smg9*) had no prior links to osteoarthritis or skeletal biology based on association with human disease and structured literature searching, whereas only two (20%) of the 10 prioritized candidates (*Scaf11*, *Smg9*) had no prior association (Fig. 1, Supplementary Data 7).

Combining findings from informatics prioritization with the seven lines with the most severe joint phenotypes (*ClusterXN1*, *Pitx1*, *Fgfbp1*, *Josd1*, *Sh3bp4*, *Unk*, *Zfp341*) identified 14 genes with strong evidence from independent sources for a functional role in osteoarthritis pathogenesis (*Pitx1*, *Bhlhe40*, *Scaf11*, *Smg9*, *Col4a2*, *Sh3bp4*, *Ccdc6*, *Herc1*, *Nebl*, *Zfp341*, *ClusterXN1*, *Fgfbp1*, *Josd1*, and *Unk*). The four genes with the strongest overall evidence were *Pitx1*, *Bhlhe40*, *Sh3bp4* and *Unk* (Fig. 1, Supplementary Data 7).

**Severe early-onset osteoarthritis in *Pitx1*+/− mice.** *Pitx1* is a homeobox transcription factor required for patterning, and specification of hindlimb morphology[29,30]. Homozygous *Pitx1*−/− mutations cause post-natal lethality with gross morphological abnormalities affecting the hindlimb skeleton[29,30]. Genomic rearrangements at the human *PITX1* locus result in homeotic arm-to-leg transformations in Liebenberg syndrome (OMIM 186550), and misexpression of *Pitx1* in the mouse forelimb recapitulates this phenotype[31]. Haploinsufficiency for *Pitx1* in mice and humans results in clubfoot and other leg malformations,

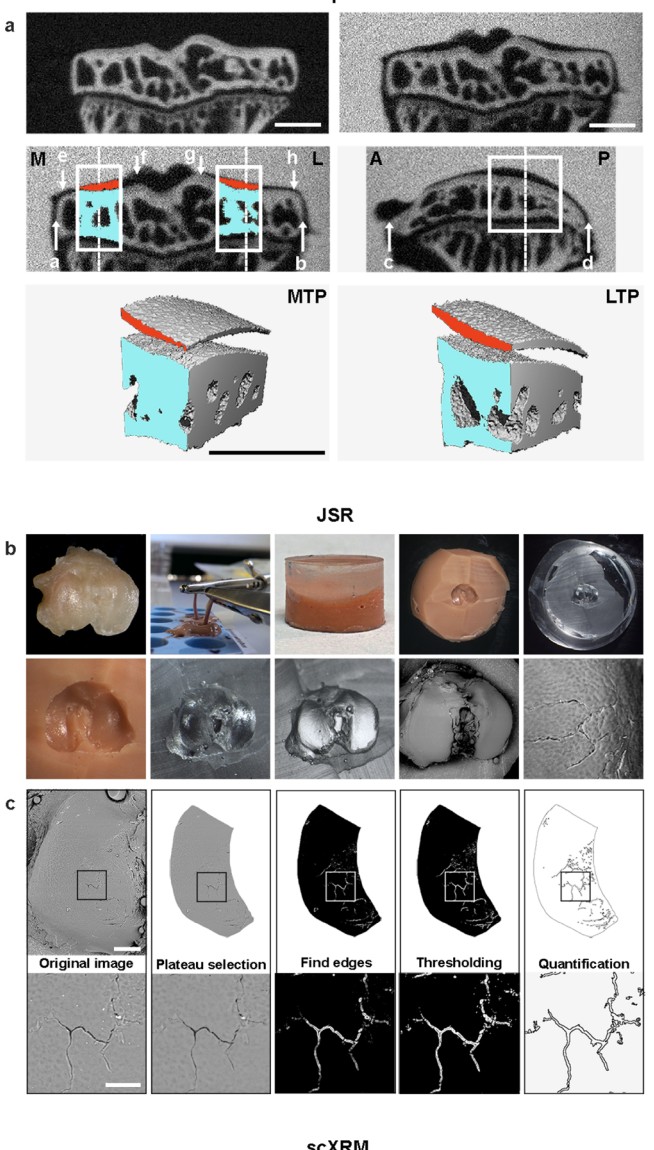

**ICEμCT**

M e f g h L    A    P

MTP    LTP

**JSR**

**scXRM**

LTP MTP    aluminium steel plastic    plastic steel aluminium
steel plastic    LTP MTP

**Fig. 2 Imaging methods. a** Coronal views of conventional μCT scan of a 16-week-old wild-type tibia (top left) and ICEμCT scan (top right), at 2 μm voxel size; mineralized tissue is white. In ICEμCT scan contrast agent at similar X-ray absorption to mineralized tissue; soft tissue including articular cartilage is black. Coronal and sagittal views of ICEμCT scans with VOIs (volumes of interest) scaled according to **a–b** (medial-lateral) and **c–d** (anterior–posterior) tibial dimensions, and positioned midway between plateau edges (medial plateau, MTP, **e–f**, lateral plateau, LTP, **g–h**, with a 7% shift medially). Plateau midpoints: dashed lines. M: medial, L: lateral, A: anterior, P: posterior. ICEμCT VOIs used for quantitative analyses (MTP, bottom left, LTP, bottom right); subchondral bone (blue); articular cartilage (red). Scale bars = 500 μm. **b** Upper images: surface of disarticulated 16-week-old wild-type mouse tibia; limbs immersed in impression medium for molding; mold (brown) overlaid with casting resin; surface of mold after removal of cast; surface of cast. Lower images: detail of mold surface; detail of cast surface; carbon-coated cast; back-scattered electron-scanning electron microscopy (BSE-SEM) image of whole tibia; high-power showing fibrillations in articular cartilage surface following DMM surgery. **c** Upper images: BSE-SEM view of lateral tibial plateau; selection of whole plateau region for analysis; automated identification of surface damage edges; thresholding of damage to capture all damage areas; damage detection. Lower images: high-power views demonstrate damage detail. Scale bars = 500 μm (upper), 100 μm (lower). **d** Upper images: Faxitron MX20; 16-week-old wild-type mouse tibia alongside aluminum (left), plastic (right) and steel (bottom) standards; plain X-ray gray scale image of tibia and standards; greyscale images are pseudocoloured according to a 16 color look-up table in which low bone mineral content (BMC) is yellow and high BMC is pink. Lower images: pseudocoloured image of proximal tibia with scaled (to width of tibia, white line) subchondral regions of interest (ROI), white boxes (LTP; lateral tibial plateau, MTP; medial tibial plateau); high-power pseudocoloured views of LTP and MTP ROIs; gray scale pixel distribution in relation to standards; gray scale distribution stretched to plastic and steel standards. Scale bar = 1 mm. *mm* millimeter, *mg HA/cm³* milligrams of hydroxyapatite/cubic centimeter.

cartilage beneath (Fig. 3, Supplementary Fig. 2c and Supplementary Data 4).

demonstrating hindlimb development is sensitive to *Pitx1* gene dosage[32].

Heterozygous *Pitx1*^{+/−} mice had the most severe phenotype observed in this study, with extensive joint damage affecting both compartments of the knee including decreased Cg.V and thickness, and increased areas of complete loss or severe erosion of articular cartilage. The *Pitx1*^{+/−} phenotype was as severe as the extensive joint damage observed in 22-week-old WT mice 12 weeks after DMM surgery (Fig. 3 and Supplementary Fig. 2a, c). Osteophytes were detected in 5/7 *Pitx1*^{+/−} mice (Supplementary Data 8). Accordingly, histology confirmed severe osteoarthritis with extensive vertical clefts in articular cartilage and areas of erosion over 50–75% of the articular surface to the calcified

**Early-onset osteoarthritis in *Bhlhe40*^{−/−} and *Sh3pb4*^{−/−} mice.** *Bhlhe40*^{−/−} mice had decreased median and maximum Cg.Th affecting both compartments of the knee. Further evidence of spontaneous damage included vertical clefts in the articular cartilage surface extending down to the layer of chondrocytes immediately below the superficial layer, together with some loss of surface lamina (Fig. 4, Supplementary Fig. 5 and Supplementary Data 4). No osteophytes were detected in *Bhlhe40*^{−/−} mice (Supplementary Data 8). *Bhlhe40* is a widely expressed transcription factor involved in regulation of cell proliferation, differentiation, apoptosis, and senescence[33]. *Bhlhe40* expression is enriched in skeletal tissues (Supplementary Data 7), specifically in proliferating and differentiating chondrocytes during endochondral ossification[34], and is elevated in response to hypoxia[35], bone morphogenetic protein-2 (BMP2) and transforming growth factor-β (TGFβ) but suppressed by parathyroid hormone[36]. *Bhlhe40* stimulates terminal chondrocyte and osteoblast differentiation[36,37], and has been implicated in bone loss in chronic periodontitis[38]. Deletion of *Bhlhe40* in mice results in increased BMC and density (Supplementary Data 7), but a role in osteoarthritis pathogenesis has not been postulated and *BHLHE40* has not been associated with osteoarthritis in GWAS.

Overall, young adult *Bhlhe40*^{−/−} mice display thinning of articular cartilage, suggesting a role in disease onset that involves the key hypoxia, BMP2, TGFβ, and PTH signaling pathways[39–42].

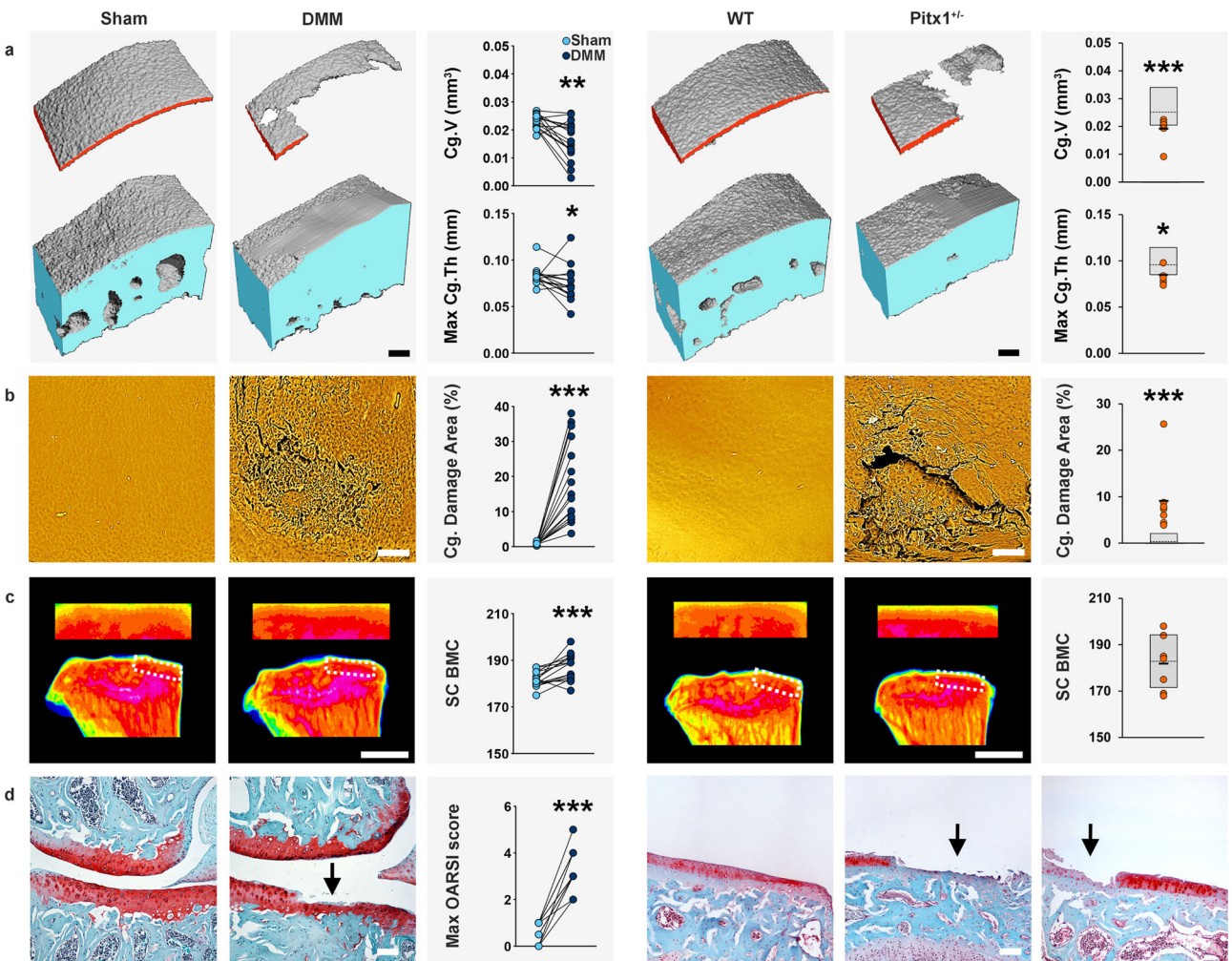

**Fig. 3 Severe early-onset osteoarthritis in *Pitx1*$^{+/-}$ mice is similar to extent of joint damage following osteoarthritis provocation surgery. a** Iodine contrast-enhanced μCT images of medial tibial plateau articular cartilage (red) and subchondral bone (blue) from 22-week-old wild-type (WT) male mice 12 weeks after sham operation or following destabilization of medial meniscus surgery (DMM; $n = 11$) to provoke osteoarthritis, and 16-week-old male WT and heterozygous *Pitx1*$^{+/-}$ mice ($n = 7$). Graphs show decreased articular cartilage volume (Cg.V) and maximum articular cartilage thickness (Max Cg. Th). **b** Back-scattered electron-scanning electron microscopy images of tibial plateau joint surface replicas from sham and DMM-operated WT mice, and WT and *Pitx1*$^{+/-}$ mice. Graphs show increased articular cartilage (Cg.) damage area. **c** X-ray microradiography images of proximal tibia and the medial tibial plateau subchondral bone region of interest (dashed box) from sham and DMM-operated WT mice, and WT and *Pitx1*$^{+/-}$ mice in which greyscale images are pseudocoloured according to a 16 color look-up table in which low bone mineral content (BMC) is yellow and high BMC is pink. Graphs show increased subchondral BMC (SC BMC; DMM only). **d** Coronal sections of knee joint compartments stained with Safranin-O/Fast green from sham and DMM-operated WT mice, and from WT and two examples of *Pitx1*$^{+/-}$ mice with extensive joint destruction. Graph shows the maximum OARSI score on the medial tibial plateau in DMM-operated mice ($n = 11$). Arrows indicate areas of cartilage destruction. Orange circles: individual mutant samples, black horizontal lines: sample mean. Gray boxes: reference ranges derived from 100 wild-type samples. For normally distributed parameters (SC BMC), reference range is two standard deviations above and below the mean (dashed line). For non-normally distributed parameters (Cg.V, Max Cg.Th, Cg. Damage Area), reference range is the 2.5th–97.5th percentile, and dashed line is the median. Scale bars = 100 μm (**a**, **b**, **d**) and 1 mm **c**. DMM vs. Sham: *$P$ < 0.05, **$P$ < 0.01, ***$P$ < 0.001, two-tailed Wilcoxon matched-pairs signed rank test for Cg.V ($P = 0.00131$), Max Cg.Th ($P = 0.017334$), Cg. Damage Area ($P = 0.000031$) and Max. OARSI score ($P = 0.000977$), and paired two-tailed t-test for SC BMC ($P = 0.000477$). *Pix1t*$^{+/-}$: *$P$ < 0.00568, ***$P$ < 0.0001, Bonferroni-corrected two-tailed Wilcoxon rank sum test. Cg.V: $P = 0.00008$, Max Cg.Th: $P = 0.00319$, Cg. Damage Area: $P = 0.00001$. mm; millimeter. Source data are provided as a Source Data file.

*Sh3bp4*$^{-/-}$ mice had decreased Cg.V and median and maximum Cg.Th, together with increased articular cartilage surface damage affecting the lateral compartment. Histology revealed moderate osteoarthritis with articular cartilage fibrillations and vertical clefts over <25% of the articular surface extending to the calcified cartilage beneath (Fig. 4, Supplementary Fig. 5 and Supplementary Data 4). No osteophytes were detected in *Sh3bp4*$^{-/-}$ mice (Supplementary Data 8). *Sh3bp4* is a poorly

characterized SH3 domain binding protein involved in transferrin receptor (TfR) internalization[43], fibroblast growth factor receptor (FGFR) trafficking[44], mammalian target of rapamycin (mTOR) signaling[45] and inhibition of the Wnt pathway[46]. *SH3BP4* has not been associated with osteoarthritis in GWAS.

In summary, *Sh3bp4*$^{-/-}$ mice display loss of articular cartilage and moderate cartilage damage. *Sh3bp4* thus represents an osteoarthritis susceptibility gene, the deletion of which accelerates

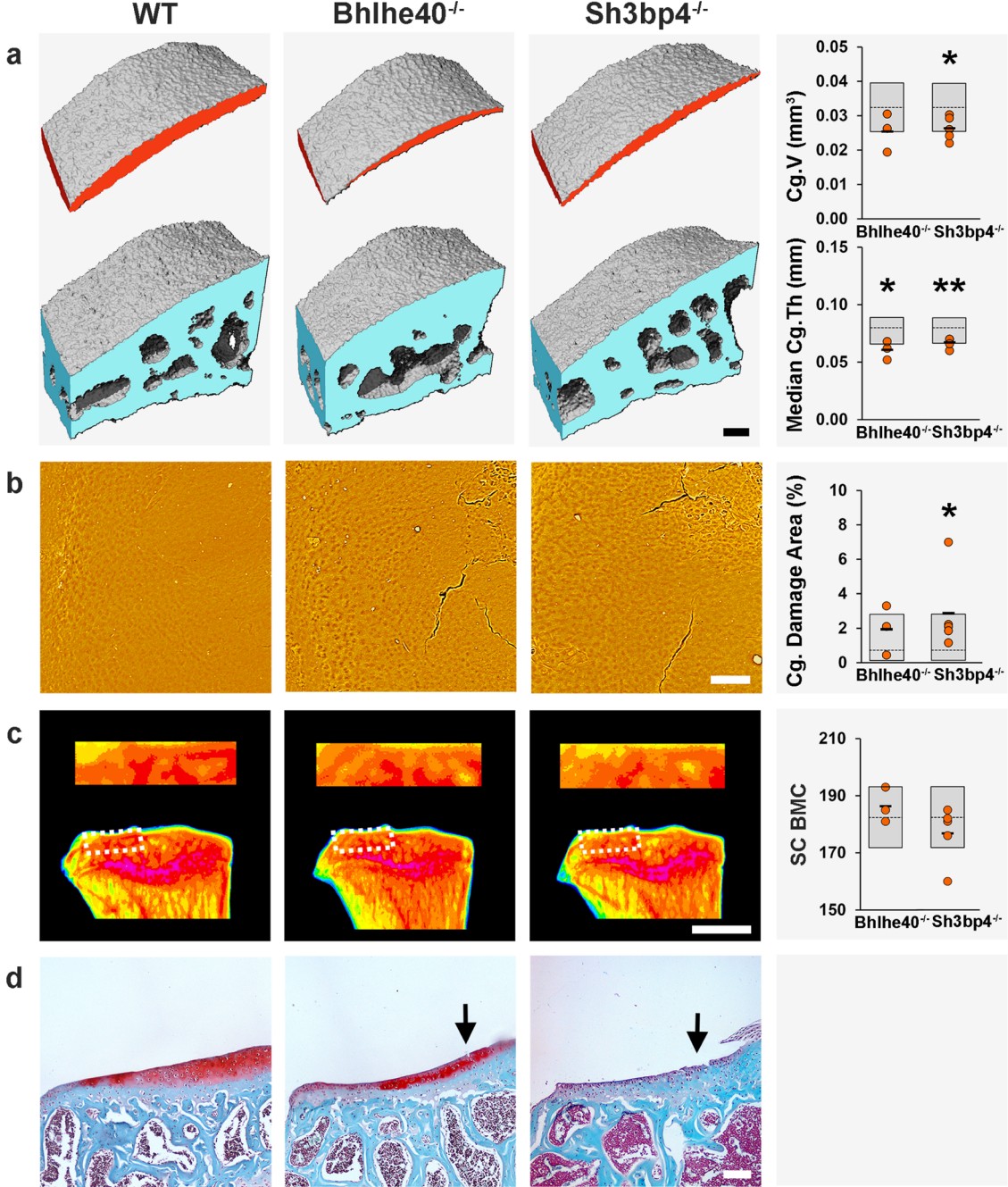

**Fig. 4 Early-onset osteoarthritis in *Bhlhe40*<sup>−/−</sup> and *Sh3pb4*<sup>−/−</sup> mice. a** Iodine contrast-enhanced μCT images of lateral tibial plateau articular cartilage (red) and subchondral bone (blue) from 16-week-old wild-type (WT) and homozygous *Bhlhe40*<sup>−/−</sup> and *Sh3bp4*<sup>−/−</sup> mice. Graphs show articular cartilage volume (Cg.V) and median articular cartilage thickness (Median Cg.Th) in *Bhlhe40*<sup>−/−</sup> ($n = 3$) and *Sh3bp4*<sup>−/−</sup> ($n = 5$) mice. **b** Back-scattered electron-scanning electron microscopy images of lateral tibial plateau joint surface replicas from WT, *Bhlhe40*<sup>−/−</sup> and *Sh3bp4*<sup>−/−</sup> mice. Graph shows articular cartilage (Cg) damage. **c** X-ray microradiography images of proximal tibia and the lateral tibial plateau subchondral bone region of interest (dashed box) from WT, *Bhlhe40*<sup>−/−</sup> and *Sh3bp4*<sup>−/−</sup> mice in which greyscale images are pseudocoloured according to a 16 color look-up table in which low bone mineral content (BMC) is yellow and high BMC is pink. Graph shows no change in subchondral BMC (SC BMC). **d** Coronal sections of lateral knee joint compartment stained with Safranin-O/Fast green from WT, *Bhlhe40*<sup>−/−</sup> and *Sh3bp4*<sup>−/−</sup> mice ($n = 3$ sections from one mouse each). Arrows indicate areas of cartilage damage. Orange circles: individual mutant samples, black horizontal lines: sample mean. Gray boxes: reference ranges derived from 100 wild-type samples. For normally distributed parameters (Cg.V, SC BMC), reference range is two standard deviations above and below the mean (dashed line). For non-normally distributed parameters (median Cg.Th, Cg. Damage Area), reference range is the 2.5–97.5th percentile, and dashed line is the median. Scale bars = 100 μm (**a, b, d**) and 1 mm (**c**). *$P < 0.00568$, **$P < 0.001$, Bonferroni-corrected two-tailed Wilcoxon rank sum test. Cg.V: $P = 0.00311$ (*Sh3bp4*<sup>−/−</sup>), Median Cg.Th: $P = 0.00469$ (*Bhlhe40*<sup>−/−</sup>), $P = 0.00081$ (*Sh3bp4*<sup>−/−</sup>), Cg. Damage Area: $P = 0.00282$ (*Sh3bp4*<sup>−/−</sup>). *mm* millimeter. Source data are provided as a Source Data file.

joint damage. Its role in disease pathogenesis may involve the key TfR, FGFR, mTOR, and Wnt signaling pathways, which regulate bone and cartilage homeostasis and tissue repair[47–50].

**Additional applications of the OBCD joint pipeline.** The development of joint phenotyping methods to investigate mutant mice demonstrates the power of unbiased functional genomics in osteoarthritis gene discovery. Open access availability of IMPC mouse lines provides a rich resource for investigation of disease mechanisms and the identification and testing of preventive or disease-modifying drugs. Here we describe three further distinct applications of the phenotyping pipeline that demonstrate how the OBCD joint phenotype database can be leveraged to add value to studies of human osteoarthritis and how the imaging techniques can be applied to address additional hypotheses.

**Knockout mice for genes differentially expressed in human OA cartilage.** Phenotyping genetically modified mouse models is a powerful method to functionally annotate potential disease susceptibility genes identified in human studies. We interrogated the OBCD database of joint phenotypes from randomly selected mouse lines to annotate the function of 409 genes differentially expressed in low- versus high-grade articular cartilage in osteoarthritis patients[51]. Phenotype data in 16-week-old mice were available from eight differentially expressed genes (*Unk*, *Josd1*, gasdermin E (*Gsdme*), Rho GTPase-activating protein 30 (*Arhgap30*), *Ccdc6*, *Col4a2*, methyl-CpG-binding domain protein 1 (*Mbd1*) and staufen double-stranded RNA binding protein-2 (*Stau2*)). Mutation of six (75%) differentially expressed genes in mice resulted in joint abnormalities whereas deletion of the other two (25%) (*Mbd1*, *Stau2*) had no effect (Supplementary Data 4 and 5). This compares with 14 out of 50 randomly selected lines (28%) with strong evidence for a functional role in osteoarthritis following prioritization in this study ($P = 0.01582$, Fisher's exact test). This enrichment of signal supports interrogation of the OBCD phenotype database to accelerate functional gene discovery in osteoarthritis. Combining complementary mouse and human gene discovery approaches demonstrates synergy, especially as none of the genes identified have been associated with osteoarthritis in GWAS.

*Unk*$^{-/-}$ mice had decreased Cg.V in the LTP and decreased maximum cartilage thickness with increased surface damage affecting the MTP (Fig. 5, Supplementary Fig. 6, Supplementary Data 4, 5, and 7). No osteophytes were detected in *Unk*$^{-/-}$ mice (Supplementary Data 8). Mean body weight was 2.49 SDs below the WT mean (Supplementary Data 4). UNK is an RNA binding zinc finger protein expressed in developing brain that controls neuronal morphology[52]. UNK is also expressed in the skeleton (Supplementary Data 7). *Unk*$^{-/-}$ mice display articular cartilage loss with early cartilage damage indicating *Unk* is a protective osteoarthritis susceptibility gene. Downregulation of UNK at the RNA and protein levels in high-grade osteoarthritis cartilage in humans suggests a role in disease pathogenesis.

*Josd1*$^{-/-}$ mice had decreased median and maximum cartilage thickness affecting both joint compartments and increased articular cartilage surface damage affecting the lateral compartment (Fig. 5, Supplementary Fig. 6, Supplementary Data 4, 5, and 7). No osteophytes were detected in *Josd1*$^{-/-}$ mice (Supplementary Data 8). JOSD1 is a widely expressed deubiquitinating enzyme that may play a role in regulation of cell membrane dynamics[53]. It stabilizes SOCS1, an important negative regulator of cytokine signaling[54,55], and *JOSD1* mRNA is upregulated in high-grade osteoarthritis cartilage (Fig. 5), suggesting a protective role for JOSD1 in osteoarthritis.

*Gsdme*$^{-/-}$ mice had increased articular cartilage surface damage affecting the lateral compartment (Fig. 5, Supplementary Fig. 6, Supplementary Data 4, 5 and 7). No osteophytes were detected in *Gsdme*$^{-/-}$ mice (Supplementary Data 8). Gasdermin E is a member of a family of proteins that facilitate necrotic programmed cell death (pyroptosis) following cleavage by caspase-3[56]. Pyroptosis has recently been shown to promote knee osteoarthritis[57,58], and upregulation of *Gsdme* in high-grade osteoarthritis cartilage (Fig. 5), supports a role for gasdermin E in disease progression.

*Arhgap30*$^{-/-}$ mice had increased articular cartilage surface damage affecting the lateral compartment and a significant outlier phenotype following Mahalanobis analysis (Supplementary Data 4 and 7). *Arhgap30* encodes a Rho GTPase implicated in cell proliferation, migration, and invasion acting via inhibition of Wnt[59]. Its putative role and increased expression in high-grade osteoarthritis cartilage (Fig. 5), suggests involvement of ARHGAP30 in cartilage repair mechanisms during osteoarthritis pathogenesis.

*Ccdc6*$^{-/-}$ mice had increased articular cartilage surface damage affecting the medial compartment (Supplementary Data 4 and 7). *Ccdc6* is a cell cycle checkpoint regulator that facilitates cell survival[60]. *CCDC6* was associated with heel BMD in a large GWAS[19] and *Ccdc6*$^{-/-}$ mice have decreased bone strength (Supplementary Data 7). Concordant with upregulation of CCDC6 in high-grade osteoarthritic cartilage (Fig. 5), *CCDC6* was differentially expressed in a meta-analysis of gene expression profiling in synovial tissue from osteoarthritis cases and controls[61]. Overall, *CCDC6* represents an osteoarthritis susceptibility gene, deletion of which may contribute to development of osteoarthritis by actions in both cartilage and the synovium.

*Col4a2*$^{-/-}$ mice had increased articular cartilage surface damage affecting the lateral compartment (Supplementary Data 4). The collagen type IV alpha 2 chain is a major component of vascular basement membrane[62], but is also present in skeletal basement membranes owing to expression in bone cells (Supplementary Data 7). *COL4A2* was differentially expressed in osteoarthritis synovium[61], in knee joints following DMM surgery[63], in a time course analysis following DMM surgery[64], and in articular cartilage biopsies from osteoarthritis patients[65]. The *COL4A2* locus was differentially methylated in a genome-wide analysis of hip compared with knee osteoarthritis cartilage[66]. In summary, several lines of evidence suggest a role for *Col4a2* in the pathogenesis of osteoarthritis and the articular cartilage damage in *Col4a2*$^{-/-}$ mice is consistent with this hypothesis (Fig. 1 and Supplementary Data 4).

**Age-related joint degeneration.** We next studied 4- and 12-month-old WT mice to investigate the effect of ageing. Joints from 12-month-old mice had features of osteoarthritis compared to 4-month-old juvenile mice. Even though articular volume and thickness did not change with age, the 12-month-old mice had significantly increased areas of cartilage damage in both lateral and medial compartments. Histology revealed an increased maximum OARSI score on the LTP in 12-month-old compared with 4-month-old mice. These changes were accompanied by loss of subchondral bone (decreased SC BV/TV, SC Tb.Th, and SC Tb.N) in the lateral compartment and increased SC BMC in the medial compartment (Fig. 6 and Supplementary Fig. 7).

***Dio2***$^{Ala92}$ **mice are protected from osteoarthritis.** We next used the phenotyping pipeline to investigate a mouse model of a common human single nucleotide polymorphism previously associated with human osteoarthritis. As proof-of-concept for functional investigation of signals arising from human genetic

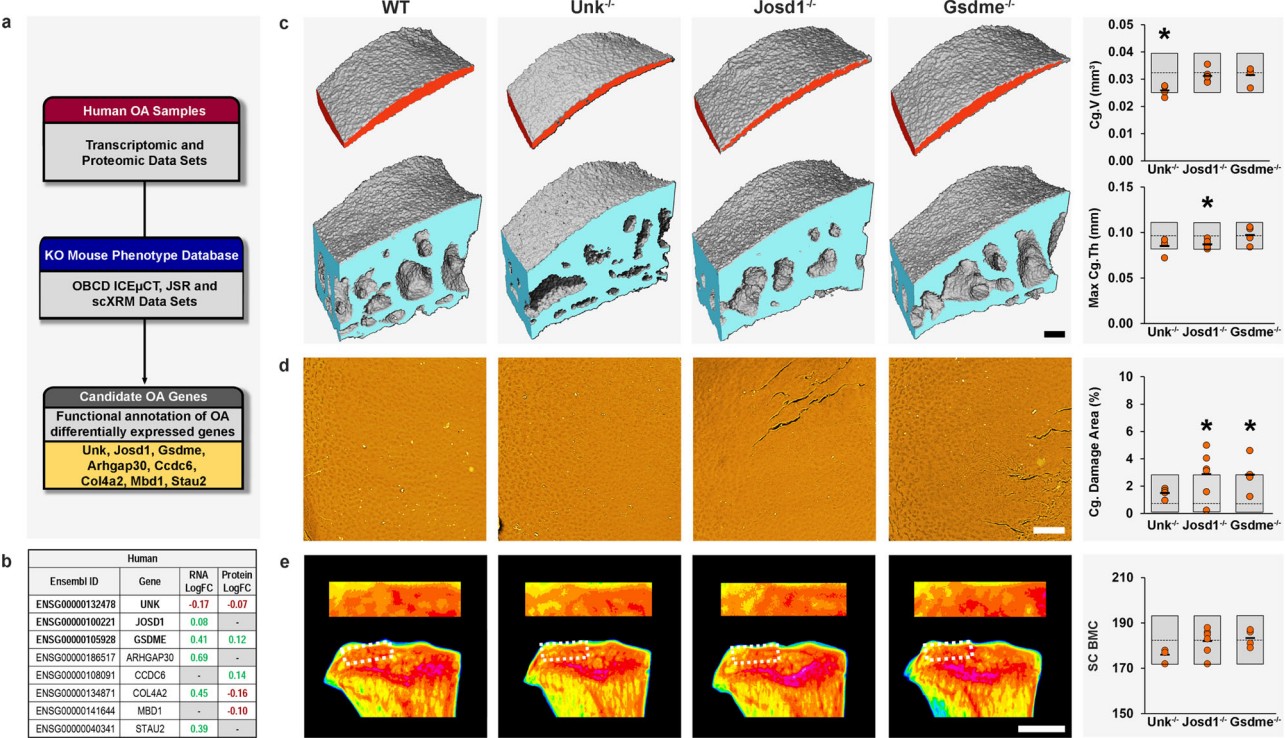

**Fig. 5 Early-onset osteoarthritis in mice with deletion of genes differentially expressed in human osteoarthritis cartilage. a** 16-week-old mice with deletion of candidate genes differentially expressed in osteoarthritic human articular cartilage (*Unk, Josd1, Gsdme, Arhgap30, Ccdc6, Col4a2, Mdb1,* and *Stau2*) were investigated for osteoarthritis phenotypes. **b** Differential expression (log fold change, logFC) of human orthologues of candidate genes in low-grade versus high-grade osteoarthritis in human articular cartilage samples. Genes with significantly increased expression in high-grade osteoarthritis cartilage shown in green and those with decreased expression in red. **c** Iodine contrast-enhanced μCT images of lateral tibial plateau articular cartilage (red) and subchondral bone (blue) from wild-type (WT), $Unk^{-/-}$, $Josd1^{-/-}$, and $Gsdme^{-/-}$ mice. Graphs show articular cartilage volume (Cg.V) and maximum articular cartilage thickness (Max Cg.Th) in $Unk^{-/-}$ ($n = 4$), $Josd1^{-/-}$ ($n = 6$), and $Gsdme^{-/-}$ ($n = 4$) mice. **d** Back-scattered electron-scanning electron microscopy images of tibial plateau joint surface replicas from WT, $Unk^{-/-}$, $Josd1^{-/-}$, and $Gsdme^{-/-}$ mice. Graph shows articular cartilage (Cg) damage. **e** X-ray microradiography images of proximal tibia and the lateral tibial plateau subchondral bone region of interest (dashed box) from WT, $Unk^{-/-}$, $Josd1^{-/-}$, and $Gsdme^{-/-}$ mice in which greyscale images are pseudocoloured according to a 16 color look-up table in which low bone mineral content (BMC) is yellow and high BMC is pink. Graph shows subchondral BMC (SC BMC Orange circles: individual mutant samples, black horizontal lines: sample mean. Gray boxes: reference ranges derived from 100 wild-type samples. For normally distributed parameters (Cg.V, Max. Cg.Th, SC BMC), reference range is two standard deviations above and below the mean (dashed line). For non-normally distributed parameters (Cg. Damage Area), reference range is the 2.5–97.5th percentile, and dashed line is the median. Scale bars = 100 μm (**c, d**) and 1 mm (**e**). *$P < 0.00568$, Bonferroni-corrected two-tailed Wilcoxon rank sum test. Cg.V: $P = 0.00215$ ($Unk^{-/-}$), Max. Cg.Th $P = 0.00362$ ($Josd1^{-/-}$), Cg. Damage Area: $P = 0.00563$ ($Josd1^{-/-}$), $P = 0.00355$ ($Gsdme^{-/-}$). *mm* millimeter. Source data are provided as a Source Data file.

association studies, we studied the effect of rs225014, a polymorphism in the human *DIO2* gene. rs225014 results in a substitution at amino acid 92 (Thr92Ala) in the DIO2 enzyme that activates thyroid hormones in target cells. The minor allele (Ala92) frequency is estimated at 40%[67]. rs225014 was positively associated with osteoarthritis in a genome-wide linkage study that included replication in a separate cohort[68], but this association was not reproduced in the Rotterdam study[69] or a later meta-analysis[70]. Nevertheless, a polymorphism in the *DIO2* promoter (rs12885300) has been associated with hip geometry in genome-wide linkage studies. Although minor alleles at rs225014 and rs12885300 were associated with opposite osteoarthritis outcomes, a haplotype combining both was associated with increased disease susceptibility[71]. Overall, these findings implicate regulation of local thyroid hormone availability in the pathogenesis of osteoarthritis and are consistent with well-known actions of thyroid hormones, which stimulate hypertrophic chondrocyte differentiation[72] and expression of cartilage matrix-degrading enzymes[73]. Nevertheless, *DIO2* has not been associated with osteoarthritis in GWAS, and the role of the *DIO2* rs225014

polymorphism remains controversial, attracting much debate even beyond the osteoarthritis field[74–76].

In vivo studies have demonstrated increased subchondral bone but intact articular cartilage in $Dio2^{-/-}$ knockout mice[77]. $Dio2^{-/-}$ mice have decreased calreticulin expression in articular cartilage[78], a gene implicated in cartilage thinning in response to mechanical loading[79], and are protected from cartilage damage following forced exercise. By contrast, increased *DIO2* expression resulted in negative effects on chondrocyte function and homeostasis in vitro[80] and *DIO2* expression was increased in articular cartilage from osteoarthritis patients[81]. Furthermore, cartilage-specific overexpression of *Dio2* in transgenic rats resulted in cartilage destruction[82]. Together, these studies suggest that decreased DIO2 expression and reduced thyroid hormone availability in the joint may protect against osteoarthritis, whereas increased DIO2 expression may increase susceptibility.

To test this hypothesis, we analyzed the joint phenotypes of $Dio2^{Thr92}$ and $Dio2^{Ala92}$ mutant mice previously generated using the *CRISPR/Cas9* system[74]. The Thr92Ala polymorphism results in decreased enzyme activity and impaired conversion of the

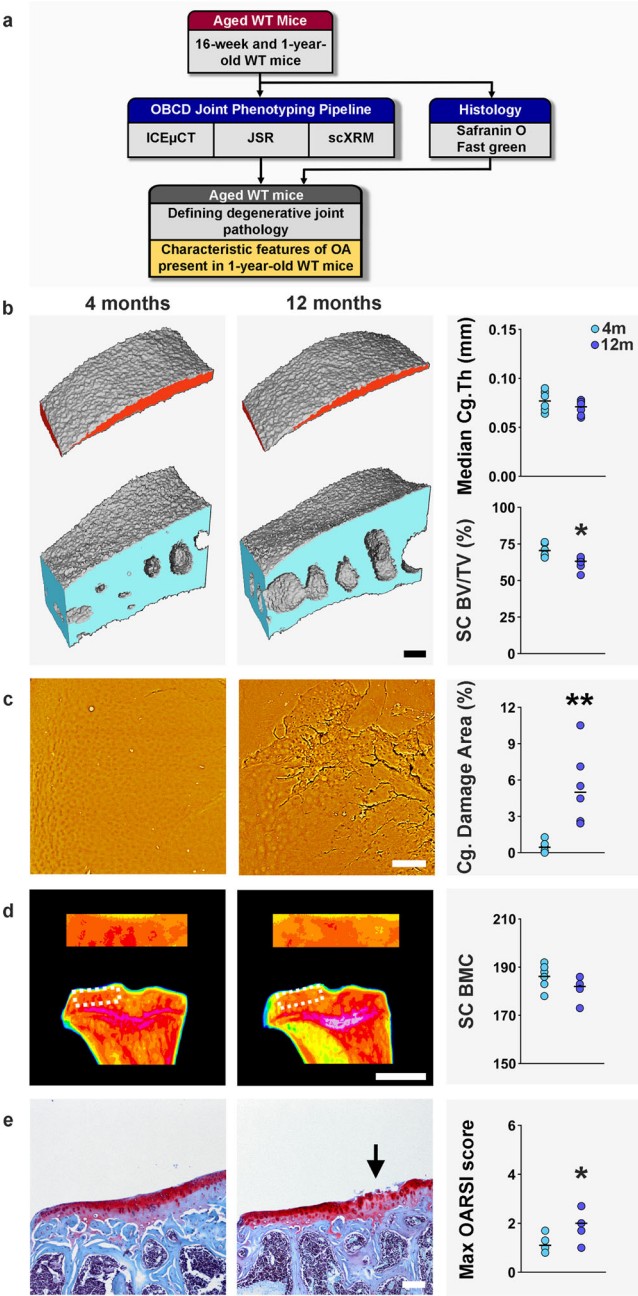

**Fig. 6 Age-related joint degeneration. a** Cohorts ($n = 6$) of young adult (4 months) and aged (12 months) wild-type (WT) mice were analyzed by iodine contrast-enhanced μCT (ICEμCT), joint surface replication (JSR) and subchondral X-ray microradiography (scXRM). **b** ICEμCT images of lateral tibial plateau articular cartilage (red) and subchondral bone (blue). Graphs show median articular cartilage thickness (Median Cg.Th) and subchondral bone BV/TV (SC BV/TV). **c** Back-scattered electron-scanning electron microscopy images of lateral tibial plateau JSRs. Graph shows articular cartilage (Cg) damage. **d** ScXRM images of proximal tibia and the lateral tibial plateau subchondral bone region of interest (dashed box) in which greyscale images are pseudocoloured according to a 16 color look-up table in which low bone mineral content (BMC) is yellow and high BMC is pink. Graph shows subchondral BMC (SC BMC). **e** Coronal sections of lateral knee joint compartment stained with Safranin-O/Fast green ($n = 6$ mice/group). Graph shows maximum OARSI histological scores. Arrow indicates cartilage damage. Scale bars = 100 μm (**b**, **c**, **e**) and 1 mm (**d**). *$P < 0.05$, **$P < 0.01$, two-tailed $t$ test (SC BV/TV, $P = 0.0082$) or two-tailed Wilcoxon rank sum test (Cg. Damage Area, $P = 0.0022$ and Max. OARSI score, $P = 0.0240$). *mm* millimeter. Source data are provided as a Source Data file.

14 genes with strong functional evidence for involvement in osteoarthritis pathogenesis. Previous approaches to mouse joint phenotyping have used modifications to standard μCT methods, including the use of cartilage staining or phase-contrast, whereas others have used confocal microscopy[83–94]. Here, we combine three complementary methods; ICEμCT, JSR, and scXRM, in the first comprehensive multiplex mouse joint phenotyping pipeline. One important consideration regarding the highly sensitive JSR method is the very low level of cartilage damage that is present in WT mice (0–2.8%). Thus, the method may be less reliable for quantitation of decreased cartilage damage in phenotype screening studies. Nevertheless, the method correlates well with OARSI histology and provides reliable and sensitive detection of increased cartilage damage, which is a pathognomonic feature of osteoarthritis. Furthermore, it is particularly useful in studies of mice with protective mutations that have been challenged by DMM provocation surgery providing such studies are adequately powered.

To demonstrate the broad utility of this joint phenotyping pipeline we describe four distinct applications (i) screening of randomly selected KO lines, (ii) analysis of knockout mice with deletion of genes differentially expressed in human OA cartilage, (iii) analysis of mice with age-related joint degeneration, and (iv) analysis of a mouse model of a common human single nucleotide polymorphism associated with osteoarthritis.

We first performed joint phenotyping in 50 randomly selected IMPC knockout lines generated by the Sanger Institute and identified one or more outlier parameters in 50% of lines. This is similar to the frequency of outlier parameters identified during bone phenotyping of randomly selected IMPC knockout mice[16,17,19] and consistent with the numerous independent loci associated with osteoarthritis[1–3] and osteoporosis[17,19] in large scale GWAS. We, therefore, developed a robust and unbiased prioritization pipeline that identified only those genes with multi-parameter joint abnormalities ($n = 14$), which were further prioritized based on biological evidence, plausibility and relevance to human disease, finally highlighting 4 out of 50 genes (*Bhlhe40, Pitx1, Sh3bp4* and *Unk*).

*Pitx1*$^{+/−}$ mice had the most severe and extensive joint destruction in these studies, with severe erosions, complete loss of articular cartilage, and osteophyte formation. Decreased *PITX1* mRNA expression in primary human articular chondrocytes, reduced PITX1 protein in histological sections of human

prohormone thyroxine (T4) to the active hormone triiodothyronine (T3), thus reducing local thyroid hormone signaling[74]. Analysis of joints from 16-week-old male mice demonstrated features of early-onset osteoarthritis in *Dio2*$^{Thr92}$ mice compared with *Dio2*$^{Ala92}$ mice. *Dio2*$^{Thr92}$ mice had decreased cartilage volume and median thickness with increased articular cartilage damage (Fig. 7 and Supplementary Fig. 8). By contrast, *Dio2*$^{Ala92}$ mutants had no signs of osteoarthritis, indicating a protective role for the Ala92 polymorphism and providing the first functional evidence of a role for this candidate *DIO2* polymorphism in vivo. The data provide further evidence that decreased thyroid hormone signaling is protective against osteoarthritis.

## Discussion

The molecular mechanisms that initiate and drive osteoarthritis progression and their genetic origins are largely unknown. We developed rapid-throughput joint phenotyping methods to identify abnormal joint phenotypes in mutant mice, identifying

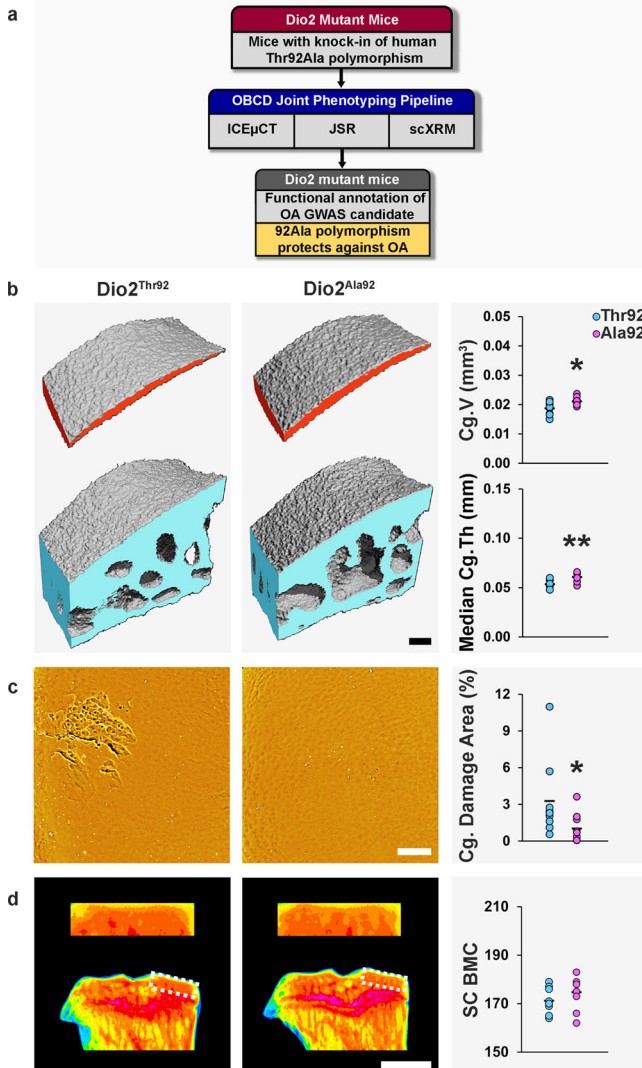

**Fig. 7 Dio2^{Ala92} mice are protected from osteoarthritis. a** 16-week-old mice with Dio2 polymorphisms homologous to the Thr92Ala polymorphism in human DIO2 (Dio2^{Thr92}, n = 9 and Dio2^{Ala92}, n = 10) were analyzed by iodine contrast-enhanced μCT (ICEμCT), joint surface replication (JSR), and subchondral X-ray microradiography (scXRM). **b** ICEμCT images of medial tibial plateau articular cartilage (red) and subchondral bone (blue). Graphs show articular cartilage volume (Cg.V) and median articular cartilage thickness (Median Cg.Th). **c** Back-scattered electron-scanning electron microscopy images of medial tibial plateau JSRs. Graph shows articular cartilage (Cg) damage. **d** ScXRM images of proximal tibia and the medial tibial plateau subchondral bone region of interest (dashed box) in which greyscale images are pseudocoloured according to a 16 color look-up table in which low bone mineral content (BMC) is yellow and high BMC is pink. Graph shows subchondral BMC (SC BMC). Scale bars = 100 μm (**b**, **c**) and 1 mm (**d**). *$P < 0.05$, **$P < 0.01$, two-tailed Wilcoxon rank sum test. Cg.V: $P = 0.019$, Median Cg.Th: $P = 0.006$, Cg. Damage area: $P = 0.016$. Source data are provided as a Source Data file.

osteoarthritic cartilage, and increased subchondral bone thickening in preliminary studies of ageing Pitx1^{+/−} mice are all consistent with an important role for PITX1 in osteoarthritis pathogenesis[95]. Overall, we show that heterozygous deletion of Pitx1, a critical developmental gene, results in severe early-onset joint damage. Pitx1 is thus an osteoarthritis gene with a key role in disease onset and progression that likely involves its effects on

joint morphology and the developmental programs that are re-initiated during disease progression.

We exploited our joint phenotype database to characterize the functions of six candidate genes differentially expressed in human osteoarthritis cartilage. These studies identify roles for Unk, Josd1, Gsdme, Arhgap30, Ccdc6, and Col4a2 in the pathogenesis of osteoarthritis. Unk^{−/−} mice were also prioritized as one of four lines with the strongest combined evidence for a key functional role in osteoarthritis pathogenesis, whereas Josd1^{−/−} mice were one of seven lines with the most severely abnormal joint phenotype (Fig. 1 and Supplementary Data 7). Overall, these findings demonstrate the value of leveraging the combined power of mouse and human experimental pipelines to enhance understanding of complex disease.

Previous histological studies have only demonstrated the onset of osteoarthritis at 15 months of age in WT C57BL6 mice[96]. The current studies, however, clearly define early features of age-related joint degeneration in 1-year old animals. Thus, our phenotyping methods are more economical and have improved sensitivity to detect early osteoarthritis compared with current labor- and resource-intensive histological techniques. OBCD joint phenotyping is one third of the cost and takes one sixth of the time compared to the current gold standard OARSI histological analysis (Supplementary Fig. 9, Supplementary Data 9–11). In addition to identifying genes that provoke early-onset of osteoarthritis when deleted, these methods now provide efficient opportunities to analyze genes that protect joints from age-related degeneration when inhibited or deleted, or for analysis of drug intervention studies.

Finally, we identified a protective role for a minor allele of the DIO2 gene (Ala92). The Ala92 variant impairs conversion of the prohormone T4 to the active hormone T3 and is controversially associated with osteoarthritis. Furthermore, it has been suggested that expression of this minor allele underlies the continuing psychological and metabolic symptoms experienced by some hypothyroid patients despite restoration of normal serum T4 levels following levothyroxine replacement[76,97]. As a result, genetic testing and treatment of hypothyroid patients with the Ala92 variant using increased doses of levothyroxine (T4) or a combination of levothyroxine and liothyronine (T3) has been proposed. Importantly, this therapeutic approach, which aims to overcome impaired conversion of T4 to T3 by the Ala92 variant, may actually have detrimental long-term consequences as our data indicate that expression of the Ala92 variant may maintain cartilage integrity. Thus, the allelic imbalance that has been reported to increase expression of the Ala92 variant in human osteoarthritis cartilage[81], may in fact represent a counter-regulatory response in damaged articular cartilage to protect against osteoarthritis progression. Our findings underscore the importance of caution and further research in this controversial area, and have important public health implications, as levothyroxine is the most commonly prescribed drug in the USA and third most commonly prescribed in the UK[98].

In these studies, we performed DMM provocation surgery in 10-week-old mice as described originally by Glasson[13,99]. Nevertheless, others now recommend undertaking DMM surgery after 12-weeks-of-age because of more advanced skeletal maturity[100]. We performed sham surgery in the contralateral knee in order to include paired control samples and minimize animal numbers in accordance with the principles of reduction, refinement and replacement (3Rs). Nevertheless, we acknowledge that sham surgery may influence joint mobility and affect phenotype severity[100]. This study identifies abnormal joint phenotypes in several knockout mice suggesting the deleted genes have important roles in joint development, maintenance and repair. However, such findings do not necessarily indicate that these genes

represent tractable drug targets in the prevention and treatment of OA. Nevertheless, we also report that *CRISPR/Cas9* targeting of a single amino acid in the *Dio2* gene identifies a role for thyroid hormone signaling in joint homeostasis and has important clinical implications.

In summary, we have shown our joint phenotyping methods have broad applications in osteoarthritis research that will accelerate functional gene discovery, advance understanding of disease pathogenesis and identify drug targets for this debilitating chronic disease.

## Methods

**Experimental models**. Mouse lines generated in this study have been deposited in the European Mutant Mouse Archive (EMMA; https://www.infrafrontier.eu/) or are available on request. Animal experiments were performed and reported in accordance with ARRIVE guidelines[101]. All animal experiments were conducted under licence at Imperial College (project licence PPL70/8785) and the Wellcome Trust Sanger Institute (project licence P77453634 and PPL80/2485) in accordance with the Animals (Scientific Procedures) Act 1986 and recommendations of the Weatherall report. Animal experiments were approved by the Sanger or Imperial College Hammersmith Campus Animal Welfare Ethical Review Bodies (AWERB) as appropriate. Studies performed at the University of Chicago were approved by the Institutional Animal Care and Use Committee (IACUC) at Rush University Medical Center (16-077 and 15-033). Male mice were used for all joint phenotype analyses.

**IMPC WT and mutant mouse strains**. Samples from 16-week old male WT and genetically modified mice designed with deletion alleles on the *C57BL/6 N Taconic; C57BL/6 N* background were generated as part of the Wellcome Trust Sanger Institute's (WTSI) Mouse Genetics Project, part of the IMPC (http://www.mousephenotype.org). Details on back-crossing status, health status, administered drug and procedures, husbandry and specific conditions (including housing, food, temperature, and cage conditions) have been reported previously[102,103] with the exception that temperature was $21 \pm 2$ °C, humidity was $55 \pm 10\%$, the diet used was Mouse Breeder Diet 5021 (Labdiet, London, UK) and the IMPReSS screen had four fewer tests (hair phenotyping, open field, hot plate, and stress-induced hypothermia). Further details available on request. Mouse strains, genotypes, and Research Resource Identifiers are included in Supplementary Data 12. Samples from male mice were used for the OBCD joint phenotyping pipeline and samples from females were used for the OBCD bone-phenotyping pipeline. The rationale that justified this approach was that (i) the gold standard DMM surgical provocation model for osteoarthritis is 100% penetrant in male mice but has variable penetrance in females[24,99], and (ii) the ovariectomy provocation model for postmenopausal osteoporosis can only be performed in females.

**WT male mice for provocation studies**. Mice were housed at Imperial College London in individually ventilated cages (Techniplast, UK) on a 12 h light/dark cycle at 20–23 °C with humidity of 45–65%. Mice were housed in sibling groups and given access to chow diet and water ad libitum. Mice were weighed weekly and were within a range of 25–35 g. Housing density was between 1 and 5 animals per cage (up to 25 g) or between 1 and 4 animals per cage (over 25 g). Health status was monitored by screening sentinel mice every 3 months, and health reports are available on request. No previous drugs, tests or procedures were administered except those specified, and no adverse events occurred. Mice were killed by exposure to increasing concentrations of $CO_2$ followed by cervical dislocation.

For surgical provocation of osteoarthritis, WT *C57BL/6* virgin male mice were purchased from Charles River Laboratories (Margate, Kent, UK; $n = 32$). Mice had been maintained as homozygous WT and back-crossed more than 20 generations (https://www.criver.com/sites/default/files/Technical%20Resources/C57BL_6%20Mouse%20Model%20Information%20Sheet.pdf). Mice were housed four per cage in sibling groups with identical enrichment and allocated to surgical groups based on cage number. Weight was monitored before and for 48 hours after surgery. Order of surgery was based on cage number and numerical mouse ID and is available on request. Surgery was performed at 10 weeks of age.

For provocation of osteoarthritis by aging, WT *129/SV/C57BL/6 J* male mice (16 weeks and 1 year of age, 12 total, six mice/group) were generated from a long-standing colony originally obtained from Jacques Samarut (École Normale Supérieure, Lyon, France)[104]. WT mice were maintained and back-crossed for >20 generations.

**Dio2 mutant mice**. Homozygous 16-week-old F3 male mice (19 mice total) in which the WT mouse deiodinase 2 (Dio2) allele at residue 92 (proline/proline) was replaced with human minor (alanine/alanine) or major (threonine/threonine) alleles by *CRISPR/Cas9* (*Dio2$^{Ala92}$*, $n = 9$ and *Dio2$^{Thr92}$*, $n = 10$) were previously generated by Applied Stemcell Inc. (Milpitas, CA, USA) and housed as described[74]. No previous drugs, tests, or procedures were administered. Information regarding back-crossing status, weight, health status, husbandry and specific conditions

(including housing, temperature, humidity, and cage conditions) is available on request.

**Human sample collection**. The study design and conduct complied with all relevant regulations regarding the use of human study participants and were conducted in accordance with the criteria set by the Declaration of Helsinki. All patients provided written, informed consent prior to participation. Tissue samples were collected from 115 patients undergoing total joint replacement surgery in four cohorts: 12 knee osteoarthritis patients (cohort 1; 2 women, 10 men, age 50–88 years, mean age 68 years); 20 knee osteoarthritis patients (cohort 2; 14 women, 6 men, age 54–82 years, mean 70 years); 13 hip osteoarthritis patients (cohort 3; eight women, five men, age 44–84 years, mean 62 years); 70 knee osteoarthritis patients (cohort 4; 42 women, 28 men, age 38-84 years, mean 70 years). Matched low-grade and high-grade cartilage samples were collected from each patient. This work was approved by Oxford NHS REC C (10/H0606/20 and 15/SC/0132). Samples from knee osteoarthritis patients were collected under Human Tissue Authority license 12182, Sheffield Musculoskeletal Biobank, University of Sheffield, UK; samples from hip osteoarthritis patients were collected under National Research Ethics approval reference 11/EE/0011, Cambridge Biomedical Research Center Human Research Tissue Bank, Cambridge University Hospitals, UK.

**Rapid-throughput OBCD joint phenotyping**

*Sample preparation.* Hindlimbs from WT and mutant mice were skinned and fixed for 24–30 h in 10% neutral buffered-formalin, rinsed twice, and stored in 70% ethanol at 4 °C. Samples were anonymized and randomly assigned to batches for rapid-throughput analysis. Prior to analysis, limbs were rehydrated in PBS + 0.02% sodium azide for >24 h. Soft tissue was removed and the knee joint was disarticulated under a Leica MZ9 dissecting microscope (Leica Microsystems, UK) with the aid of fine forceps (Dumont #5, Cat#11252-20; Fine Science Tools, Germany) and 5 mm spring scissors (Vannas Tubingen, Cat#15003-08, Fine Science Tools, Germany).

*Iodine contrast-enhanced μCT.* This method was developed to detect signs of osteoarthritis, including cartilage damage and subchondral bone sclerosis or bone loss. The joint is immersed in an iodinated contrast agent with similar X-ray absorption to bone and imaged (Fig. 2). This allows segmentation of both the articular cartilage and underlying subchondral bone in the same volume of interest. The technique can also be used to analyze the condyles and head of the femur.

Rehydrated tibiae were blotted on low-linting tissue (Kimtech, UK) to remove all liquid. Single tibiae were placed into 6 mm sample holders (Scanco Medical AG, Brüttisellen, Switzerland) in 180 mg/mL ethiodized oil (Lipiodol Ultra, Guerbet Laboratories, Solihull, UK) mixed with sunflower oil. The tibial epiphysis (1–1.5 mm) was scanned using a Scanco μCT-50 micro-computerized tomography (μCT) scanner with X-ray tube potential 70 kVp, 200 μA tube current, 2 μm voxel size with 1500 ms integration time and 1× averaging. Documented quality control scans were performed routinely as indicated by the manufacturer and conform to reporting recommendations[105]. Images were processed using the Scanco software (μCT Tomography v6.4-2/Open VMS), with Gaussian filtration (standard deviation, sigma; 0.8, support; (1), and component labeling for the 20 (subchondral bone) and 5 (articular cartilage) largest particles. A three-voxel (subchondral bone) and five-voxel (articular cartilage) dilation/erosion was applied.

Analysis was performed using Xming 6.9.0.31 (©2005-2007 Colin Harrison) software implemented with the pUTTy client program (release 0.62 ©1997-2011 Simon Tatham). Tomographs were rotated twice to place the limb in a standard position. First, the tibia was rotated clockwise to point the anterior-most patellar surface vertically. Second, the tibia was rotated 90° to place the medial-lateral axis of the tibia facing the viewer (Fig. 2a, coronal views). Medial and lateral tibial plateaux were analyzed separately. Equally sized medial and lateral VOIs were scaled automatically according to the size of the tibia (Fig. 2a, medial-lateral dimensions, a–b; anterior–posterior dimensions, c–d). The width of each VOI was equal to 14% of the medial-lateral axis, which equated to 59% of the medial plateau width and 53% of the lateral plateau width. The anterior–posterior dimension of each VOI was 33% of the size of each tibia. VOIs were located at the same positions on each limb by determining the plateau edges (Fig. 2a, e–h). Each VOI was then centered on the medial-lateral midpoint of the plateau (Fig. 2a, dashed line), and the lateral plateau VOI was further offset by a distance equivalent to 7% of the medial-lateral axis (Fig. 2a). The medial VOI midpoint was 61.5%, and the lateral VOI midpoint 68.5% (7% posterior shift) along the anterior–posterior axis. This accounted for the different shapes of the medial and lateral plateau, based on standard measurements determined from mean values obtained from 10 WT mice to capture the thickest part of each plateau (Supplementary Data 13).

For articular cartilage, all tissue below a density threshold of 200 mg hydroxyapatite/cubic centimeter (mg HA/cm$^3$) was analyzed. The volume and thickness (median and maximum) were determined. For subchondral bone, all tissue between the articular cartilage and the growth plate above a density threshold of 300 mg HA/cm$^3$ was analyzed. Relative SC BV/TV, TMD, Tb.Th, Tb.N) were determined. Of note, the use of the Lipiodol contrast agent in ICEμCT results in a systematic and predictable increase in subchondral bone parameters (Supplementary Data 13). However, since all samples are processed and imaged

identically this systematic increase does not compromise data interpretation as results represent relative comparisons.

For osteophytes, coronal sections located at the midpoint of the VOI were scored for the presence or absence of osteophytes by three blinded independent scorers (Supplementary Fig. 3, Supplementary Data 2, 8).

To validate ICEµCT, we compared the maximum cartilage thickness (Cg.Th) determined from OARSI histological sections with the same parameter determined by ICEµCT. We measured Cg.Th in ICEµCT images and histological sections from 23 samples that had undergone OBCD rapid-throughput joint phenotyping and histological analysis (samples obtained from sham-operated ($n = 3$) and DMM-operated ($n = 3$) 12 weeks after surgery at 22 weeks-of-age, WT mice at 16- ($n = 7$) and 52- ($n = 6$) weeks-of-age, $Sh3bp4^{-/-}$ ($n = 1$), $Pitx1^{-/-}$ ($n = 2$), $Bhlhe40^{-/-}$ ($n = 1$) at 16 weeks-of-age). Linear regression was performed to determine the correlation between the two methods and Bland-Altman plots were generated[83,106]. Maximum Cg.Th determined by the two techniques was strongly correlated ($R^2 = 0.91$, MTP, $P < 0.0001$, $R^2 = 0.755$, LTP, $P < 0.0001$, Supplementary Fig. 10). Furthermore, there was no proportional bias in measurements, and no increase in differences between methods with phenotype severity (Bland-Altman plot; MTP; bias, $-0.0002273$ mm, $R^2 = 0.125$, $P = 0.106$, LTP; bias, $0.004217$ mm, $R^2 = 0.011$, $P = 0.63$, Supplementary Fig. 10).

Nevertheless, the LTP Max Cg.Th was significantly thicker when measured by histology (mean value $+4.4\%$) when compared with measurement by ICEµCT (*$P = 0.014$) (Supplementary Fig. 10e). To investigate this difference, we tested whether ICEµCT sample preparation resulted in shrinkage. However, ICEµCT analysis of fresh and fixed samples showed no significant difference in cartilage volume and thickness (Supplementary Fig. 10f). Overall, these studies demonstrate an absence of shrinkage during ICEµCT sample preparation, and a consistent difference in Max Cg.Th observed in the LTP but not the MTP when comparing ICEµCT with histology. These findings suggest a difference in anatomical orientation of the LTP relative to the MTP that is revealed when 2D (histology) and 3D (ICEµCT) imaging are compared. We hypothesize that the apparent increase in LTP thickness determined by histology results from a horizontal anterior–posterior tilt in the LTP relative to the MTP, but this relative anatomical relationship does not affect the Max Cg.Th measured in three dimensions by ICEµCT.

*Joint surface replication*. This method was developed to determine articular cartilage surface damage. The technique enables visualization of joint surface morphology at high-resolution in the natural hydrated condition by using a resin cast as a surrogate (Fig. 2b, c). The method can also be used to analyze the condyles and head of the femur.

*Molding joint surface replicas*. Disarticulated, rehydrated tibiae were positioned with the articular surfaces parallel to the molding surface using Draper Helping Hand brackets with articulated arms (31324Hh, Bamford Trading, Ross-on-Wye, UK). A multi-aperture molding template of depth >0.5 cm was custom-made. Moisture was removed to preserve surface detail by blotting with low-linting tissue (Kimtech). Samples were immersed immediately in Virtual Light Body dental impression medium (Ivoclar-Vivadent, Leicester, UK) for 4 minutes prior to de-molding and immediate return to storage medium. Molds were visually inspected for defects or bubbles and either cast immediately or stored for a maximum of 2 weeks according to manufacturer's advice.

*Casting joint surface replicas*. Joint surface replicas were cast using Crystal Clear 202 acrylic resin (Smooth-On, Bentley Advanced Materials, London, UK) according to manufacturer's instructions (Fig. 2). Pre-warmed (37 °C) parts A and B (1:9 ratio) were mixed thoroughly and vacuum de-gassed. Resin was poured into joint surface molds by gravity flow using a fine pipette tip with the aid of a dissecting microscope (MZ9, Leica Microsystems, UK) and bubbles were removed from mold-resin interfaces. Casts were set for 16 h at 18–23 °C, post cured for 6 h at 65 °C and removed from molds after 7 days at room temperature, according to manufacturer's instructions. Each cast was secured to a custom-produced $10 \times 10 \times 0.3$ cm aluminum raft with double-sided carbon tape (Agar Scientific, Stansted, UK). Rafts of samples were coated with >20 nm carbon using a high-vacuum bench-top carbon coater (Agar Scientific).

*Imaging joint surface replicas*. Coated samples were analyzed using a Vega3 XMU scanning electron microscope (Tescan UK, Cambridge) at high vacuum with a four-quadrant back-scattered electron detector (Deben, Bury St. Edmunds, UK). Tibial plateaux were imaged at $1536 \times 1536$ pixel image size with a scan speed of 5 and $3\times$ frame-corrected averaging over a view field of $1800 \times 1800$ µm (Fig. 2). Beam voltage was 20 kV and working distance was 17 mm. ImageSnapper semi-automated imaging software (V1.0, Tescan UK) was used to acquire images. The presence or absence of osteophytes was determined by three blinded independent scorers (Supplementary Fig. 3, Supplementary Data 2, 8).

*Damage quantitation in joint surface replicas*. A quantitation pipeline was developed to determine areas of damage on the joint surfaces of the medial and lateral plateau accurately, whereas excluding extraneous surface debris (Fig. 2c). Macros (ImageJ1.44) were produced for automated analysis (see below; Supplementary Data 14). Samples were analyzed in batches by a blinded operator. After each step,

macros included batch-saving of modified images, unless otherwise stated. Particle sizes are based on pixel size of 1.37 µm². Steps for semi-automated damage quantitation were:

1. Select plateau (*free select* tool; Fig. 2c, second panel and close-up). Isolated plateaux were batch-saved with a modified file name (Macro 1; automated).
2. Exclude bright outlier particles <4 pixels (5.5 µm²; Macro 2; automated).
3. Detect damage using the *find edges* tool (Macro 3, Fig. 2c, third panel and close-up; automated).
4. Manually erase debris (soft tissue, bubbles in molded surfaces) with close comparison to the original image and batch save with a modified file name (Macro 4; automated).
5. Adjust the threshold to capture all damage accurately with close comparison with the original image and batch save with a modified file name (Macro 5, Fig. 2c, fourth panel and close-up; automated).
6. Quantify damage area (*analyze particles* tool; Macro 6, Fig. 2c, fifth panel and close-up) particle size >20 pixels (27.5 µm²), circularity 0–0.5 (0; line; 1; perfect circle). Excluding circular objects further refines the damage selection as circular objects are not likely to be genuine damage (automated). Damage area was recorded and represented as percentage of the total plateau area. The whole plateau area was determined by performing steps 1–2 above, and then:
7. Set threshold to minimum and batch save with a modified file name (Macro 5; automated).
8. Determine whole plateau area (*analyze particles* tool; Macro 6; plateau outline in Fig. 2c, 5th panel), particle size >100 pixels (188 µm²), circularity 0–1 (automated).

*Subchondral X-ray microradiography*. Quantitative scXRM was performed at 10 µm pixel resolution using a Faxitron MX20 variable kV point-projection X-ray source and digital image system (Qados, Cross Technologies plc, Berkshire, UK) with modifications to previously published protocols[21,22]. The protocol is detailed in full below. Disarticulated tibiae were imaged flat with the posterior surface facing downwards (Fig. 2d). Samples were imaged at 26 kV for 15 s with the sample tray raised 6 mm above the ×5 magnification slot to maximize analysis area. Quantitation was performed in ImageJ1.44 by stretching pixel information between density values for plastic (minimum) and steel (maximum) standards and assigning to one of 256 bins using Macro 7 and automated batch saving (Supplementary Data 14). The width of each tibia at the growth plate was determined and used to automatically scale the regions of interest (ROIs; height: 9%, width: 34%, of the growth plate width). These dimensions were based on an optimal ROI size determined from 10 WT tibiae that was sufficiently wide to include the whole subchondral plate and shallow enough to avoid calcified cartilage. The ROI positions were parallel to each plateau and included the topmost yellow pixel representing the bone directly beneath the tibial plateau. For each ROI, the number of pixels in each bin was determined using the *Custom Histogram* macro (http://rsb.info.nih.gov/ij/). The median gray level (density bin at which the cumulative frequency of pixels reaches 50%) defines the relative BMC. BMC determined by scXRM correlated with BMC determined by ICEµCT (tissue mineral density multiplied by bone volume (TMD*BV) in 100 WT mice; LTP ($R^2 = 0.601$, ***$P = 2.77 \times 10^{-11}$) and MTP ($R^2 = 0.531$, ***$P = 2.77 \times 10^{-11}$; Supplementary Data 6).

*Repeatability and precision*. To determine the repeatability of each method, seven WT samples were selected that were evenly distributed across the reference range. Samples were blinded and analyzed five times in a random order, and in a different random order for each method. Mean, standard error of the mean, absolute precision error (standard deviation; PE(SD)), and precision error as percentage of coefficient of variation (PE(%CV))[107] were calculated for each sample across five measurements (Supplementary Fig. 1b and Supplementary Data 15). Results were also expressed as the number of standard deviations away from the reference range mean (Supplementary Data 15). The average precision error for each parameter was calculated using the root-mean-square[107] (Supplementary Data 15).

To determine whether the precision error of the methods increases with phenotype severity in mutants, we performed an additional repeatability analysis on a subset of 6 knockout mouse lines with a wide range of phenotype severities ($Bhlhe40^{-/-}$, $Gsdme^{-/-}$, $Josd1^{-/-}$, $Pitx1^{+/-}$, $Sh3bp4^{-/-}$, and $Unk^{-/-}$). Samples were analyzed in an identical manner to the WT cohort described above. Spearman correlations were calculated comparing phenotype severity with absolute precision error [PE(SD)] and with precision error as a percentage coefficient of variation [PE (%CV)]. Linear regression analysis demonstrated that precision error did not change with phenotype severity for 17/18 parameters. Nevertheless, the absolute precision error of MTP cartilage surface damage [Cg. Damage Area (%)] did increase with phenotype severity ($R^2 = 0.906$, $P = 0.001$; Supplementary Data 16).

To assess repeatability of the methods in both WT and mutant cohorts, two-way mixed-model intra-class correlation coefficients (single measure, absolute agreement) with 95% confidence intervals were calculated for repeated analyses by a single rater[83,108]. All parameters had intra-class correlation coefficients >0.8, indicating excellent reproducibility (Supplementary Data 15).

*Surgical provocation of osteoarthritis.* Mice underwent DMM[13,24] osteoarthritis provocation surgery at 10 weeks-of-age. To minimize recovery time, mice were anaesthetized using inhaled isofluorane/oxygen mix (1–1.5 L/min), and breathing rate was monitored visually throughout surgery. The medial meniscal-tibial ligament (MMTL) of the right knee was transected using a 5 mm microsurgical blade (World Precision Instruments, Hitchin, UK, Cat# 500249). Sham surgery was performed on the left knee, in which skin and joint capsule were opened but the MMTL was left intact. Each joint capsule was closed with a single suture, and the skin was closed using intradermal sutures. Temperature was maintained using a heat-pad and monitored by rectal thermometer. Post-operative buprenorphine (0.1 mg/kg in saline, Vetergesic; Ceva United Kingdom, Amersham, UK), and carprofen (5 mg/kg in saline, Rimadyl; Zoetis, Leatherhead, UK) were administered subcutaneously post surgery while under anesthetic, and daily for 48 h. Dosage was calculated based on individual weight. Physical pain indicators (including mobility, piloerection, and hunching) were monitored post surgery and daily for 48 h. Surgeries were performed between 10 am and 4 pm in a designated surgical suite. Mice were returned to the home cage for recovery. Mice were housed four per cage in sibling groups and sacrificed 12 weeks post surgery.

Left and right hindlimbs from half of the experimental cohort (16 mice; 2 mice/cage) were phenotyped by rapid-throughput OBCD joint phenotyping. The limbs of the other half of the surgical cohort (2 mice/cage) were sectioned and scored for osteoarthritis using gold standard protocols. Numbers of mice in each group were based on group numbers in published guidelines[24].

*Histological scoring of DMM surgical samples.* Articulated left (sham) and right (DMM) hindlimbs were decalcified for 7–10 days in 10% ethylenediaminetetraacetic acid and embedded in paraffin wax in a 90° flexed physiological position. Coronal 4 μm sections were cut at 80 μm intervals through the knee joint and stained with Safranin-O/Fast green and Weigert's haematoxylin counterstain[77,109]. Sections from five levels within the articulating zone centered around the midpoint were scored for each limb. All sections were scored by three blinded observers using the OARSI scoring system[23]. Summed and maximum scores were calculated for each compartment (medial tibial plateau; MTP, medial femoral condyle; MFC, lateral tibial plateau; LTP, lateral femoral condyle; LFC) in a single limb from five sections. In cases when five sections were not available, scores were weighted according to the following equation:

$$\text{Weighted score} = \text{score} + [(5 - N) \times (\text{score}/N)] \tag{1}$$

where $N$ was the number of scoreable sections. Weighted scores were rounded to the nearest 0.5. Sum scores represent the sum of scores from five levels. The maximum score is the highest score (out of a maximum of six). The total joint sum score is the sum of scores for all compartments across five levels. The total joint maximum score represents the maximum score from all compartments across five levels. The presence of osteophytes was also scored (Supplementary Data 1). Lack of displacement of the medial meniscus was considered a criterion for exclusion ($n = 1$ mouse). Synovitis severity was scored in the same sections [(severity score = sum of pannus formation score (0–3) + synovial lining hyperplasia score (0–3) + sub-synovial inflammation score (0–3)][110–112]. Five sections from each knee were scored by a blinded scorer. Differences between mean scores were evaluated by two-tailed Wilcoxon matched-pairs signed rank tests (Supplementary Fig. 3, Supplementary Data 17).

*Analysis of osteophytes.* We scored osteophytes by histology, ICEμCT and JSR in sham and DMM-operated limbs 12 weeks after surgery at 22 weeks-of-age, and by ICEμCT and JSR in mutant mouse lines at 16 weeks-of-age. In sham and DMM-operated limbs, the numbers of samples in which osteophytes were detected by ICEμCT and JSR did not differ from the number of samples in which osteophytes were detected by histology ($P > 0.9999$; $P = 0.7874$, Fisher's exact test; Supplementary Data 2), indicating detection of osteophytes by ICEμCT and JSR is equivalent to detection by OARSI histology. A combination of ICEμCT and JSR is required for osteophyte analysis in mutant mice, as JSR may not detect osteophytes in dysmorphic or severely damaged joints (Supplementary Data 8).

*Co-registration with gold standard histology.* To co-register data obtained from OBCD joint phenotype analysis directly with OARSI histological scoring, we performed further histology and OARSI scoring of samples from three mice that had previously been phenotyped in the OBCD pipeline and found to have mild, intermediate and severe degrees of joint damage (Supplementary Fig. 4). In all, 4 μm coronal sections were cut at 50 μm intervals through the disarticulated tibiae and femora. Five sections centered around the midpoint of each femur and tibia were analyzed by three independent scorers, and the mean summed and maximum scores determined.

*Functional genomics analysis of human cartilage.* We isolated chondrocytes from each cartilage sample as described[113] and detailed in full in Supplementary Methods (see Supplementary Information). Assays and analyses were performed as described in Supplementary Methods (see Supplementary Information), and RNA sequencing was performed on the Illumina HiSeq2000 or Hiseq4000 (75 bp paired-ends), with quality control including FastQC 0.11.5 (http://www.bioinformatics.

babraham.ac.uk/projects/fastqc). For raw RNA sequencing data details see Data Availability Statement. For protein extracts, we carried out digestion, 6-plex or 10-plex tandem mass tag labeling and peptide fractionation. For samples from 12 knee osteoarthritis patients, we applied a liquid chromatography mass spectrometry (LC-MS) analysis using the Dionex Ultimate 3000 ultra-high-performance liquid chromatography (UHPLC) system coupled with the high-resolution LTQ Orbitrap Velos mass spectrometer (Thermo Fisher Scientific GmbH, Dreieich, Germany). For all remaining samples, LC-MS analysis was performed on the Dionex Ultimate 3000 UHPLC system coupled with the Orbitrap Fusion Tribrid Mass Spectrometer (Thermo Fisher Scientific). For proteomics data details see Data Availability Statement.

*OBCD bone phenotyping.* Samples from 16-week-old female IMPC knockout mice were used to determine bone mass and strength parameters in the rapid-throughput OBCD bone-phenotyping pipeline[16,17,19], and results are reported in Supplementary Data 7. Samples were stored in 70% ethanol, anonymized and randomly assigned to batches for rapid-throughput analysis. The relative BMC and length of the femur and caudal vertebrae were determined at 10 μm pixel resolution by digital X-ray micro-radiography (Faxitron MX20). Micro-CT (Scanco μCT-50, 70 kV, 200 μA, 0.5 mm aluminum filter) was used to determine cortical bone parameters (thickness, BMD, internal diameter) at 10 μm voxel resolution in a 1.5 mm region centered on the mid-shaft region 56% of the way along the length of the femur distal to the femoral head, and trabecular bone parameters (bone volume, trabecular number, thickness, spacing) at 5 μm voxel resolution in a 1 mm region beginning 100 μm proximal to the distal growth plate. Biomechanical variables of bone strength and toughness (yield load, maximum load, fracture load, energy dissipated prior to fracture) were derived from destructive three-point bend testing of the femur and compression testing of caudal vertebrae 6 and 7 (Instron 5543 load frame, 100 N and 500 N load cells). Overall, 19 skeletal parameters were reported for each individual mouse studied and compared to reference data obtained from 320 16-week-old WT female mice of identical genetic background.

*Prioritization pipeline.* A prioritization pipeline was developed based on the severity of the mutant mouse phenotype and (i) additional skeletal abnormalities identified in mutant mice, (ii) gene expression in skeletal cells and tissues, (iii) gene association with human monogenic and complex diseases, and (iv) structured literature searching (Supplementary Data 7).

Genes were allocated overall priority scores out of a total maximum of 21 as follows:

A maximum score of six for rapid-throughput OBCD joint phenotyping (outlier according to criteria of reference range, Wilcoxon test followed by Bonferroni adjustment and Mahalanobis analysis, one point each; abnormal articular cartilage morphology, articular cartilage damage, subchondral bone morphology, one point each).

A maximum score of five for abnormal skeletal phenotypes reported in genetically modified mice by other studies. The following resources were searched: (1) OBCD bone phenotype data, (2) International Mouse Phenotyping Resource of Standardized Screens (IMPReSS; https://www.mousephenotype.org/impress), (3) Deciphering the Mechanisms of Developmental Disorders database (DMDD; https://dmdd.org.uk/), (4) PubMed (https://www.ncbi.nlm.nih.gov/pubmed/) using the search criteria "GENE NAME AND (knockout OR deletion OR mutation)", and (5) Mouse Genome Informatics database (http://www.informatics.jax.org/). The presence of an abnormal skeletal phenotype was assigned 1 point per database.

A maximum score of four for expression in skeletal tissues and cells (the skeleton, chondrocytes, osteoblasts/osteocytes, and osteoclasts; Supplementary Data 7). Expression in the skeleton (one point) was determined by searching MGI (http://www.informatics.jax.org/) and BioGPS (http://biogps.org/#goto=welcome). An expression level greater than the median was considered to indicate tissue or cell type expression. Expression in chondrocytes (one point) was determined by searching published transcriptome data[27] and SkeletalVis[28] (http://phenome.manchester.ac.uk/). Search was limited by species (human) in both datasets. In Kean et al.[27] a baseMean expression greater than the median was considered to indicate cell type expression. In SkeletalVis searches, a fold change > 2 ($P < 0.001$) was considered to indicate cell type expression. Expression in osteoblasts and/or osteocytes (one point) was determined by searching BioGPS, SkeletalVis, and osteocyte RNAseq data[19]. Expression in osteoclasts (one point) was determined by searching BioGPS, human osteoclast RNAseq data sets, and SkeletalVis. A baseMean expression greater than the median was considered to indicate cell type expression.

A maximum score of four for association with monogenic and polygenic skeletal disease (Supplementary Data 7). Monogenic disease association was determined by searching MGI for human-mouse disease association, and Online Mendelian Inheritance in Man (OMIM, https://www.omim.org/) by gene name (one point). Polygenic disease association was assessed by searching the European Bioinformatics Institute Genome-wide Association Studies database (EBI GWAS Catalog, https://www.ebi.ac.uk/gwas/) for association with arthritis and skeletal diseases (one point each).

A maximum score of two for publications related to skeletal tissues and cells (Supplementary Data 7). Systematic structured searches of PubMed and Google Scholar (https://scholar.google.co.uk/) were performed using the search string "GENE NAME AND (arthritis OR cartilage OR skeleton OR chondrocyte OR

osteoblast OR osteocyte OR osteoclast)". For PubMed results, the following scores were assigned: 0 (<1 publication), 0.5 (1–24 publications), or 1 (>25 publications). For Google Scholar results, the following scores were assigned: 0 (<100 results), 0.5 (100–999 results), or 1 (>1000 results).

*Quantitation and statistical analysis.* Statistical details and results of experiments are included in the relevant results section, figure legends, Reporting Summary, and below. For all experiments, *n* refers to biological replicates (number of mice or histological sections). Measurements were taken from distinct samples unless otherwise indicated. Exact *P* values are reported in figure legends and Supplementary Data files.

*OBCD joint phenotyping.* Reference range data were calculated from 100 WT mice for each parameter, and the mean, standard deviation, median, and percentiles determined (Supplementary Data 3). Frequency distribution of data sets was assessed by the Shapiro–Wilk normality test. Reference ranges 2 standard deviation above or below the mean (normally distributed parameters), or between the 2.5–97.5th percentiles (non-normally distributed parameters), were determined. An outlier was defined if the mean value of a phenotype parameter was outside the defined reference range.

Normally distributed parameters were articular cartilage volume (LTP), maximum articular cartilage thickness (LTP), SC BV/TV (LTP), subchondral trabecular thickness (LTP, MTP), subchondral trabecular number (LTP), subchondral tissue mineral density (LTP), and subchondral bone mineral content (LTP, MTP). Non-normally distributed parameters were articular cartilage volume (MTP), median articular cartilage thickness (LTP, MTP), maximum articular cartilage thickness (MTP), articular cartilage surface damage (LTP, MTP), subchondral bone volume/tissue volume (MTP), subchondral trabecular number (MTP), and subchondral tissue mineral density (MTP).

To determine the statistical significance of outlier parameters identified relative to the reference range, we performed non-parametric two-tailed Wilcoxon rank sum tests (Supplementary Data 5, *wilcox.test* function in R). For each parameter, this analysis calculated whether the distribution parameters in mutant samples was significantly different to the distribution of parameters in the 100-sample WT reference range. To account for multiple testing when defining statistical significance, we calculated the effective number of tests and then applied a Bonferroni correction as follows. Because several of the 18 phenotype parameters determined in the 100 WT mice used to calculate reference data were correlated (Supplementary Data 6, *$P$ < 0.05), we first calculated the number of effective tests using WT data in R. We calculated the pairwise correlations between all 18 parameters and obtained eigenvalues of the correlation matrix using the *eigen* function in R. We then calculated the number of effective tests ($N_{eff}$) as:

$$N_{eff} = N - \sum_\lambda I(\lambda > 1)^*(\lambda - 1) \qquad (2)$$

where $N = 18$ is the number of parameters, and $\lambda$ denotes the eigenvalues, resulting in an effective $N_{eff} \leq 8.8$. Thus, applying a Bonferroni correction for the effective number of tests, statistical significance for the differences between WT reference data and results from each mutant mouse line was $P < 0.00568$, with a second more stringent threshold of $P < 0.0001$ also correcting for the number of mouse lines. For each of 18 parameters across the 50 mouse lines, differences between mutant and WT values were determined by two-tailed Wilcoxon rank sum tests.

Differences between young and old WT mice, and between $Dio2^{Thr92}$ and $Dio2^{Ala92}$ mice, were determined using unpaired two-tailed *t* tests or two-tailed Wilcoxon rank sum tests.

Differences between sham and DMM-operated limbs from the same mouse were determined using paired two-tailed *t* tests, or two-tailed Wilcoxon matched-pairs signed rank tests.

Differences between OARSI histology and synovitis scores in sections from sham and DMM-operated knee joints were determined using paired two-tailed Wilcoxon matched-pairs signed rank tests. Differences between OARSI histology scores of sections from young and aged WT mice were determined using unpaired two-tailed Wilcoxon rank sum tests.

*Power calculations.* To determine the sample size required, we determined coefficients of variation (CV, standard deviation/mean) for normally distributed parameters, and percentage median absolute deviation from the median (median absolute deviation from the median (MAD)/median) for non-normally distributed parameters (Supplementary Data 3). The effect size (*d*) was set at twice the standard deviation (*σ*) or half of the 95% range as a percentage of the median. To detect this effect size with 80% power, the number of mice per group (*N*) was determined by

$$N = 2\frac{(Z_{1-\alpha/2} + Z_{1-\beta})^2 \times \sigma^2}{d^2} \qquad (3)$$

where power index $(Z_{1-\alpha/2} + Z_{1-\beta})^2$ is 7.85 for a power of 80%, with MAD replacing σ for non-normally distributed parameters. Thus, four mice are required to detect an effect size of two standard deviations at 80% power, and six mice at 95% power, for all parameters. For non-parametric data, 1–4 mice are required, depending on the parameter, to detect an outlier phenotype lying outside the 95% confidence intervals of the reference range (Supplementary Data 3). Minimum

detectable effect sizes for all parameters at 80%, 90%, and 95% power are presented in Supplementary Data 18.

*Mahalanobis analysis.* Robust Mahalanobis distances[114,115] were determined to identify outliers in multivariate data. These distances measure how far each observation is from the center of a data cluster, taking into account the variances of the variables and the covariances of pairs of variables. Robust Mahalanobis distances ($MD_i$)

$$MD_i = \sqrt{(x_i - T(X))C(X)^{-1}(x_i - T(X))^T} \qquad (4)$$

were calculated for each sample (represented by a vector of multivariate observations $x_i$). T(X) is a robust (i.e., relative unaffected by outliers) estimate of the mean vector and C(X) is a robust estimate of the covariance matrix. Under the assumption of multivariate normality, the distribution of $MD_i^2$ is approximately chi-squared with *p* degrees of freedom (where *p* is the number of variables). This means that any observation $x_i$ with $MD_i^2 > \chi_{p;\,0.975}^2$ can be considered an outlier. The assumption of multivariate normality was checked by plotting the ordered distances against the corresponding quantiles of an appropriate chi-squared distribution. Robust estimates of the mean and covariance matrix are used so that potential outliers are not masked. The masking effect, by which outliers do not necessarily have a large Mahalanobis distance, can be caused by a cluster of outliers that attract the mean and inflate the covariance in its direction. By using a robust estimate of the sample mean and covariance, the influence of these outliers is removed and the Mahalanobis distance is able to expose all outliers. The minimum volume ellipsoid method was used to calculate robust estimates of the mean and covariance matrix. Given *n* observations and *p* variables, the minimum volume ellipsoid method seeks an ellipsoid containing

$$h = \left[\frac{(n+p+1)}{2}\right] \qquad (5)$$

*h* points of minimum volume. All multivariate analysis was conducted in R (R Project for Statistical Computing). Overall, a mouse line was defined to have an abnormal joint phenotype using this method only when 50% or more individual samples from that line were identified as outliers following Mahalanobis analysis.

*Analysis of RNA and protein in human tissue.* Full details of this analysis are described in the Supplementary Methods (see Supplementary Information). In brief, gene-level RNA quantification was carried out using Salmon 0.8.2[116] and txiimport[117]. After quality control, we retained expression estimates for 15,249 genes with counts per million of 1 or higher in at least 40 cartilage samples (combining low-grade and high-grade), and matched low-grade and high-grade cartilage samples from 83 patients. To identify gene expression differences between high-grade and low-grade cartilage, we carried out multiple analyses using limma[118,119], DESeq2[120], and edgeR[121,122] with five different designs to account for potential confounders. This yielded 2557 genes with significant differential expression between low-grade and high-grade cartilage at 5% false discovery rate (FDR) across all five analysis designs and all five testing methods.

To carry out protein identification and quantification, we submitted the mass spectra to SequestHT search in Proteome Discoverer 2.1 and searched all spectra against a UniProt fasta file that contained 20,165 reviewed human entries. After quality control, we retained 4801 proteins that were quantified in at least 30% of samples, and matched low-grade and high-grade cartilage samples from 99 patients. To account for protein loading, abundance values were normalized by the sum of all protein abundances in a given sample, then log2-transformed and quantile normalized. An analysis for differential abundance was carried out using limma[118] with 2016 proteins showing significant differences at 5% FDR.

We used Ensembl38p10 to identify human orthologues for 50 mouse genes phenotyped by the OBCD joint phenotyping pipeline. We identified high-confidence one-to-one orthologues for 43/50 genes.

**Reporting summary.** Further information on research design is available in the Nature Research Reporting Summary linked to this article.

## Data availability
All data sets generated and/or analyzed during the current study are included in Supplementary and Source Data, and available from the corresponding authors on reasonable request or as detailed below. Figures 1, 3–7, and Supplementary Figs. 1–8 and 10 have associated raw data (included in the Supplementary Data and Source Data). In addition, raw RNA sequencing data have been deposited in the European Genome-Phenome Archive [EGA; https://ega-archive.org/) with the identifiers EGAS00001002255, EGAD00001003355 (*n* = 17), EGAD00001003354 (*n* = 9), EGAD00001001331 (*n* = 12). Proteomics data have been deposited in the PRoteomics IDEntifications database (PRIDE; https://www.ebi.ac.uk/pride/archive/) with the identifiers PXD006673, PXD002014, and PDX014666. Public databases used in this study are: BioGPS (http://biogps.org/#goto=welcome), Deciphering the Mechanisms of Developmental Disorders database (DMDD; https://dmdd.org.uk/), European Mutant Mouse Archive (EMMA; https://www.infrafrontier.eu/), European Bioinformatics Institute Genome-wide Association Studies database (EBI GWAS

Catalog, https://www.ebi.ac.uk/gwas/), Google Scholar (https://scholar.google.co.uk/), International Mouse Phenotyping Resource of Standardized Screens (IMPReSS; https://www.mousephenotype.org/impress), Mouse Genome Informatics database (http://www.informatics.jax.org/), Online Mendelian Inheritance in Man (OMIM, https://www.omim.org/), PubMed (https://www.ncbi.nlm.nih.gov/pubmed/), and SkeletalVis28 (http://phenome.manchester.ac.uk/). There are no restrictions on data availability. Source data are provided with this paper.

## Code availability

Code generated during this project is provided as Supplementary Data 14 and in the public repository Github (https://github.com/Molendo/OBCD). Repository: Molendo. Project: OBCD (search term: Molendo/OBCD).

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

## Acknowledgements

We thank Jayashree Bagchi-Chakraborty and Mahrokh Nodani for technical assistance and Carmen Huesa for assistance with synovitis scoring. We thank members of Sanger Mouse Pipelines (Mouse Informatics, Molecular Technologies, Genome Engineering Technologies, Mouse Production Team, Mouse Phenotyping) and the Research Support Facility for provision and management of mice. This work was funded by a Wellcome Trust Strategic Award (101123). Generation of mutant mice was funded by the Wellcome Trust (098051). G.R.W., J.H.D.B. were funded by a Wellcome Trust Joint Investigator Award (110140, 110141) and European Commission Horizon 2020 Grant (666869), E.Z. by the Wellcome Trust (206194), and PIC by Mrs. Janice Gibson and the Ernest Heine Family Foundation.

## Author contributions

A.B., E.Z., P.I.C., G.R.W., and J.H.D.B. conceived and designed experiments, N.C.B., V.D.L., J.G.L., J.A.W., A.B., G.R.W., and J.H.D.B. developed methods, N.C.B., J.G.L., and J.H.D.B. wrote software code, N.C.B., J.S., R.J., G.R.W., and J.H.D.B. performed statistical analyses, N.C.B., K.F.C., H.D., D.K.E., N.S.M., A.A., V.D.L., J.G.L., J.A.W., E.G., L.S., and E.A.M. conducted experiments, J.S., L.S., S.E.Y., J.M.W., V.E.V., F.K., J.K.W., D.J.A., C.J.L., A.C.B., P.I.C., G.R.W., and J.H.D.B. provided experimental resources, V.D.L. curated data, A.B., E.Z., P.I.C., G.R.W., and J.H.D.B. acquired funding, N.C.B., G.R.W., and J.H.D.B. produced figures, N.C.B., G.R.W., and J.H.D.B. wrote the manuscript, and all authors reviewed and edited the manuscript.

## Competing interests

The authors declare no competing interests.
