## [Peer Review File · Nature Communications]

Reviewers' Comments:

Reviewer #1:

Remarks to the Author:

In this manuscript, Butterfield et al report using a unique set of imaging modalities to identify joint phenotypes in mice relevant to osteoarthritis. The goal of the work was to develop "rapid-throughput" quantitative imaging so that multiple mutant strains of mice could be screened to identify new genes involved in OA as well as to test candidate genes, which they did with a risk variant in *Dio2*. The strength of the study is that the genes selected from phenotypes identified by imaging mouse knees from the International Knockout Mouse Consortium were prioritized by overlapping the results with additional extensive data from gene expression results in OA cartilage, human disease associations and a literature search. Another strength was that both a surgically-induced OA model (DMM) and natural age-related OA were examined. Several new genes associated with OA were identified. However, there were some weaknesses that need to be addressed in order better justify the conclusions of the study.

1. It was not clear how the 50 lines for the original analysis were chosen. The term "unselected mutant lines" is confusing. Was this meant as randomly selected mutant lines?
2. To judge if the rapid-throughput technique is truly rapid, the authors should include some estimate of the time taken to complete the imaging studies. Preparation and scanning times as well as time required to analyze the imaging results were not provided. Also a cost estimate per mouse would be helpful. Expensive techniques were used that included microCTs and scanning EM.
3. The study focused on two features of OA: cartilage damage/loss and subchondral bone changes. Osteophytes and synovitis are two other features that are quite important in human OA that were not evaluated. Some data on both of these features would be available from the histology studies and also osteophytes from the microCTs and should be included to better characterize the leading candidate genes.
4. It appears that almost all of the studies were performed with male mice which is a limitation that should be addressed in the discussion. Some studies were done in female mice but it was not exactly clear what these were used for. A statement early in the methods where the mice were described should state that male mice were used for all studies "except_fill in when used_".
5. There are other limitations that need to be included in a limitation paragraph in the discussion including performing DMM surgery on 10 week-old mice that are not yet skeletally mature (a minimum of 12 weeks is now the recommendation) and performing sham surgery in the contralateral knee which alters joint mobility. Another limitation is that the study focused exclusively on mice with gene deletions (except for the *Dio2* mice). Identifying an OA phenotype in mice with a deletion of a specific gene indicates that gene is important to normal joint development and/or homeostasis but does not necessarily mean that it is a good target for OA and in particular if it is a "drugable" target.
6. When reading the results it was hard to determine which age mice were being reported in all sections. For example, on page 5 "validation of new imaging techniques" and page 6 "definition of WT reference ranges". In Fig 7 results it is also not clear what age is studied to determine protection from age-related OA. Since age is an important factor for OA development the results should provide ages in each section.
7. The reporting of OARSI scores in the figures is not consistent. OARSI scores are reported as sum scores in Fig 3 but switch to max scores in Fig 6. In Fig 3 no OARSI scores are given for the *Pitx1* mice and in Fig 4 OARSI scores are missing. All figures should be consistent. Also in each figure where histology is used as an outcome the OARSI scores should be provided and as noted above osteophyte and synovial scores should be added.
8. In Figure 2 the legend mentions that conventional CT is shown on the top left but the figure has it on the top right with ICE on the top left. Also the bottom panel "c" should be "d" since C was already used.

Reviewer #2:

Remarks to the Author:

In this manuscript, the authors develop, validate and use a novel rapid-throughput imaging techniques to identify abnormal joint phenotypes in unselected mutant mice. Using the technique, they identify 14 genes with functional involvement in osteoarthritis pathogenesis, and functionally characterize 6 candidate human osteoarthritis genes in mouse models.

While the results are very interesting and massive, I still would have liked to see some more rigor in the establishment of the imaging protocol. Typically in imaging, people thrive to establish accuracy, precision, and sensitivity of the technique prior to the application of the protocol. While there is evidence for precision and some effort in the establishment of sensitivity, I was not able to see results related to the assessment of accuracy of the different parameters.

1. Accuracy is typically assessed by comparing the newly established morphometric parameters to a gold standard measure, e.g. cartilage thickness as assessed from micro-CT could be compared to thickness measurements in histological sections. Such analysis is often accompanied by Bland-Altman plots to establish the bias in the analysis between two different techniques. A good example of such an approach can be found in Stok et al, PLoS One. 2016 Jan 25;11(1):e0147564. doi: 10.1371/journal.pone.0147564. In that article, the authors also proposed nomenclature for cartilage morphometric indices that are in accordance with the standardized bone morphometry nomenclature that you are also using, which makes it is easier to compare cartilage and bone morphometric results. For example, I would recommend to use Cg.Th rather than AC Th, similar to the use of Tb.Th in your trabecular bone area.

2. The authors have undertaken a rigorous assessment of reproducibility but I would appreciate if averages for the precision errors could be given rather than simply %CV per animal. As the errors always have to be compared to the variance in the population, intra-class correlations are helpful to understand how reproducible the results really are. This also points to another problem with the reproducibility analysis as only "normal" animals have been assessed for the precision study. It is likely though that the outliers will be more difficult to assess and reproducibility will be worse in those cases. Experiments should be added to establish average precision errors in the mouse lines with aberrant phenotypes.

3. Finally, I am not fully convinced about the sensitivity analysis as only differences in aging animals to detect disease onset as well as surgically provoked late-stage disease are reported, which is nice to see but what you really want to see is how good you are to see phenotypic differences in the different genetic lines. Typically, people have used a strategy to try to establish a measure of a minimally detectable difference. I understand that you are only interested in extreme outliers but I think a more comprehensive analysis of how small a difference you are able to detect with your method would be valuable.

4. There are a number of published papers that have used that rigor for the establishment of new morphometric measures for mouse phenotyping, which could be consulted and referenced for the establishment of your imaging protocol. This is not the first time a rapid-throughput pipeline for mouse skeletal phenotyping has been established, although I must say that it is very nice to see how you multiplex different imaging techniques.

5. Please report all imaging and image processing related settings as laid out in the small animal micro-CT imaging guideline paper by Bouxsein et al, J Bone Miner Res. 2010 Jul;25(7):1468-86. doi: 10.1002/jbmr.141. This would include x-ray tube energy, tube current, frame averaging settings, image filtration settings, etc.

6. Please describe how you measured SC BMD. Is that the bone mineral in the TV or in the BV? Also, it is not clear to me why you are using scXRM to define BMC. As you are already measuring BMD with micro-CT, BMC is simply BMD times TV, or BV depending on your normalization. Also, it would be good to show that these numbers are consistent, which means that correlation of

BMD*TV or BMD*BV should be highly correlated. Is that the case?

7. When reading through your introduction as a representative of the general readership of your manuscript, it is not clear to me how you chose the 50 unselected lines. I find it quite astonishing that you found an OA phenotype in 50% of the lines. Would you expect to find even more relevant genes in the 1000s of other lines that would still need to be phenotyped? If that is the case, I am not so sure why it is important to discuss potential implications of these genes in such detail but solely based on literature findings without any mechanistic insight. It will necessary to help the reader better understand your rational for the choice of lines.

8. Along those lines, it is not clear to me why you chose to generate a Crispr/Cas9 mutant mice with a Thr92Ala polymorphism in the Dio2 gene. Wouldn't it have been more interesting to generate mutate mice along the pathways of the genes that you have identified in your phenotyping approach to demonstrate proof-of-concept?

9. One limitation is that you only focused on male mice. No rational was given. Do your findings hold true for female mice as well? By the way, on page 36, under Skeletal Phenotyping, you say female mice. I guess this is a typo. Otherwise, the text would have to be adapted in the results section.

10. You mention that "Details on backcrossing status, weight, health status, administered drug and procedures, husbandry and specific conditions (including housing, food, temperature and cage conditions) have been reported previously (88,89) ...". First, I think it is maybe a little much to ask the reader to find this data in these other publications, and second, you should mention whether there are any significant differences between the lines in any these parameters but especially for weight, which might explain differences in the arthritic phenotype as well. Some of these measures may be added to Supplementary Table 1 and discussed in the results section together with the other differences.

11. You mentioned that this is rapid-throughput imaging approach. Too me, I cannot see a lot of automation in your image analysis approach and therefore would have to conclude that your approach is very labor intensive. You should give numbers of how much time is needed for one sample to be phenotyped for each of the techniques as well as how much time it took from receiving a batch of samples until you had the final results.

12. On page 29, you mention "data not shown". I thought that current good practice has changed and that all data mentioned needs to be shown or referenced. Otherwise, the text should be omitted.

13. On page 30, you introduce two thresholds in mg/hydroxyapatite/cm³ to either delineate cartilage or bone, respectively. What is the influence of the contrast agent on the choice of threshold?

14. Please label all Tables in the supplementary part. Now there are only the table titles but not the actual table number.

Ralph Müller

Reviewer #3:

Remarks to the Author:

In accordance with the Editor's request, this reviewer provides a review with aspects related to CRISPR. I leave the evaluation of other aspects of the manuscript to other referees.

In the context of CRISPR gene editing, Thr92 and Ala92 mutant mice were used in this study. This reviewer found that these mice have previously been created and validated (ref. #74). Since some of the authors of #74 are identical with that of this manuscript, not the newly created mice but the previously established ones were supposedly used in this study. Therefore, almost no methodological and scientific description was found in the manuscript concerning CRISPR gene editing. I just provide a few suggestions to improve the manuscript.

Specific comments:

1. If this reviewer's understanding is correct, Thr92 and Ala92 mutant mice used in this study were already reported previously. However, there are many inappropriate description throughout the manuscript. For example, in Abstract, the authors described: "we generate mutant mice with an osteoarthritis-associated polymorphism in the Dio2 gene by Crispr/Cas9 genome editing and demonstrate..." To avoid misunderstanding, the sentence should be corrected, for example - "we used the previously established mutant mice..." Similarly, the whole manuscript should be revised accordingly. If the knock-in mice were newly created in this study, thorough description of the creation and validation of mutant mice must be added.
2. In accordance with the description in #74, the wild-type gene of mouse Dio2 contains Pro92, but such explanation was not found in the current manuscript. The authors should add some more background regarding Dio2 gene.
3. Both "CRISPR" and "Crispr" were found in the manuscript. The terminology should be unified as "CRISPR."

POINT-BY-POINT RESPONSE TO REVIEWERS' COMMENTS

Reviewers' comments:

Reviewer #1 (Remarks to the Author):

In this manuscript, Butterfield et al report using a unique set of imaging modalities to identify joint phenotypes in mice relevant to osteoarthritis. The goal of the work was to develop “rapid-throughput” quantitative imaging so that multiple mutant strains of mice could be screened to identify new genes involved in OA as well as to test candidate genes, which they did with a risk variant in Dio2. The strength of the study is that the genes selected from phenotypes identified by imaging mouse knees from the International Knockout Mouse Consortium were prioritized by overlapping the results with additional extensive data from gene expression results in OA cartilage, human disease associations and a literature search. Another strength was that both a surgically-induced OA model (DMM) and natural age-related OA were examined. Several new genes associated with OA were identified. However, there were some weaknesses that need to be addressed in order better justify the conclusions of the study.

1. It was not clear how the 50 lines for the original analysis were chosen. The term “unselected mutant lines” is confusing. Was this meant as randomly selected mutant lines?

> We agree and have replaced “unselected mutant lines” with “randomly selected mutant lines” throughout the manuscript, as requested.

2. To judge if the rapid-throughput technique is truly rapid, the authors should include some estimate of the time taken to complete the imaging studies. Preparation and scanning times as well as time required to analyze the imaging results were not provided. Also a cost estimate per mouse would be helpful. Expensive techniques were used that included microCTs and scanning EM.

> We thank the reviewer for this excellent suggestion and we now include a comprehensive time and cost analysis of the OBCD rapid-throughput techniques in comparison to the current gold-standard OARSI histology analysis. Full details of these analyses are included in Supplementary Figure 9 and Supplementary Tables 9-11. In summary, OBCD joint phenotyping is one third of the cost and takes one sixth of the time compared to OARSI histology (page 19, lines 466-468).

3. The study focused on two features of OA: cartilage damage/loss and subchondral bone changes. Osteophytes and synovitis are two other features that are quite important in human OA that were not evaluated. Some data on both of these features would be available from the histology studies and also osteophytes from the microCTs and should be included to better characterize the leading candidate genes.

> We agree with the reviewer and now include data on both osteophytes and synovitis, as follows:

Osteophyte analysis

Data are presented in Supplementary Figure 3 and Supplementary Tables 1, 2, 8

Results are described on page 6, lines 132-136; page 11, line 253; page 11, lines 263-264; page 12, lines 283-284; page 13, lines 325-326; page 14, lines 336-337; page 14, lines 344-345; page 18, line 443

Analysis methods are described on page 32, lines 805-807; page 34, lines 858-860; page 39, lines 991-999

Synovitis analysis

Data are presented in Supplementary Figure 3 and Supplementary Tables 1, 16

Results are described on page 6, lines 132-136

Analysis methods are described on page 39, lines 984-988

4. *It appears that almost all of the studies were performed with male mice which is a limitation that should be addressed in the discussion. Some studies were done in female mice but it was not exactly clear what these were used for. A statement early in the methods where the mice were described should state that male mice were used for all studies “except_fill in when used_”.*

> Thank you, we agree this was not explicit and confirm all joint phenotyping was performed in males, clarifying this on page 28, lines 688-689 and lines 702-707.

5. *There are other limitations that need to be included in a limitation paragraph in the discussion including performing DMM surgery on 10 week-old mice that are not yet skeletally mature (a minimum of 12 weeks is now the recommendation) and performing sham surgery in the contralateral knee which alters joint mobility. Another limitation is that the study focused exclusively on mice with gene deletions (except for the Dio2 mice). Identifying an OA phenotype in mice with a deletion of a specific gene indicates that gene is important to normal joint development and/or homeostasis but does not necessarily mean that it is a good target for OA and in particular if it is a “drugable” target.*

> We thank the reviewer for noting these limitations and now include discussion of these issues as requested on page 20, lines 491-502.

6. *When reading the results it was hard to determine which age mice were being reported in all sections. For example, on page 5 “validation of new imaging techniques” and page 6 “definition of WT reference ranges”. In Fig 7 results it is also not clear what age is studied to determine protection from age-related OA. Since age is an important factor for OA development the results should provide ages in each section.*

> We are grateful to the reviewer for noting this and agree. We have now stated the age of the mice in the results and figure legends throughout the manuscript to provide clarification.

7. *The reporting of OARSI scores in the figures is not consistent. OARSI scores are reported as sum scores in Fig 3 but switch to max scores in Fig 6. In Fig 3 no OARSI scores are given for the Pitx1 mice and in Fig 4 OARSI scores are missing. All figures should be consistent. Also in each figure where histology is used as an outcome the OARSI scores should be provided and as noted above osteophyte and synovial scores should be added.*

> We agree and now include the maximum OARSI score consistently in Figures 3 and 6. Although histology images are shown for Pitx1 mice in Figure 3 and Bhlhe40 and Sh3bp4 mice in Figure 4, these are images of representative histological sections only as there are insufficient sample numbers available for formal OARSI scoring of IMPC mice screened through the OBCD joint pipeline. Nevertheless, our validation studies in DMM operated mice demonstrated excellent concordance between OBCD phenotype analysis and OARSI histological scoring (see Supplementary Figures 2, 3, 4, 10). Furthermore, osteophyte and synovitis data are now presented in Supplementary Figure 3 and Supplementary Tables 1, 2, 8, 16 with details of text references in the response to point 3 above.

8. *In Figure 2 the legend mentions that conventional CT is shown on the top left but the figure has it on the top right with ICE on the top left. Also the bottom panel “c” should be “d” since C was already used.*

> We thank the reviewer for pointing out our error and have amended Figure 2 accordingly.

Reviewer #2 (Remarks to the Author):

In this manuscript, the authors develop, validate and use a novel rapid-throughput imaging techniques to identify abnormal joint phenotypes in unselected mutant mice.

Using the technique, they identify 14 genes with functional involvement in osteoarthritis pathogenesis, and functionally characterize 6 candidate human osteoarthritis genes in mouse models.

While the results are very interesting and massive, I still would have liked to see some more rigor in the establishment of the imaging protocol. Typically in imaging, people thrive to establish accuracy, precision, and sensitivity of the technique prior to the application of the protocol. While there is evidence for precision and some effort in the establishment of sensitivity, I was not able to see results related to the assessment of accuracy of the different parameters.

1. Accuracy is typically assessed by comparing the newly established morphometric parameters to a gold standard measure, e.g. cartilage thickness as assessed from micro-CT could be compared to thickness measurements in histological sections. Such analysis is often accompanied by Bland-Altman plots to establish the bias in the analysis between two different techniques. A good example of such an approach can be found in Stok et al, PLoS One. 2016 Jan 25;11(1):e0147564. doi: 10.1371/journal.pone.0147564. In that article, the authors also proposed nomenclature for cartilage morphometric indices that are in accordance with the standardized bone morphometry nomenclature that you are also using, which makes it is easier to compare cartilage and bone morphometric results. For example, I would recommend to use Cg.Th rather than AC Th, similar to the use of Tb.Th in your trabecular bone area.

> We thank the reviewer for these excellent suggestions and have now included all the analyses requested as follows. Cartilage thickness analysis and Bland-Altman plots are presented in Supplementary Figure 10 and raw data is included in Source Data files. Analytical methods are described on pages 32-33, lines 809-821. Furthermore, we have changed the cartilage and bone morphometric nomenclature to the terminology recommended throughout the manuscript.

2. The authors have undertaken a rigorous assessment of reproducibility but I would appreciate if averages for the precision errors could be given rather than simply %CV

per animal. As the errors always have to be compared to the variance in the population, intra-class correlations are helpful to understand how reproducible the results really are. This also points to another problem with the reproducibility analysis as only “normal” animals have been assessed for the precision study. It is likely though that the outliers will be more difficult to assess and reproducibility will be worse in those cases. Experiments should be added to establish average precision errors in the mouse lines with aberrant phenotypes.

> We thank the reviewer for highlighting this important point and have now included all requested analyses as follows. The results from new repeatability and precision analyses are provided in Supplementary Figure 1 and Supplementary Tables 14 and 15, and the methods are described on pages 36-37, lines 912-938.

3. Finally, I am not fully convinced about the sensitivity analysis as only differences in aging animals to detect disease onset as well as surgically provoked late-stage disease are reported, which is nice to see but what you really want to see is how good you are to see phenotypic differences in the different genetic lines. Typically, people have used a strategy to try to establish a measure of a minimally detectable difference. I understand that you are only interested in extreme outliers but I think a more comprehensive analysis of how small a difference you are able to detect with your method would be valuable.

> We agree with the reviewer that such analyses provide valuable additional information. We have performed power calculations to determine minimally detectable differences and include these new data in Supplementary Table 17. Details of power calculations are described in the methods section on page 45, lines 1143-1158.

4. There are a number of published papers that have used that rigor for the establishment of new morphometric measures for mouse phenotyping, which could be consulted and referenced for the establishment of your imaging protocol. This is not the first time a rapid-throughput pipeline for mouse skeletal phenotyping has been

established, although I must say that it is very nice to see how you multiplex different imaging techniques.

> We thank the reviewer for noting our omission and have now cited new references and added text in the discussion on page 18, lines 434-440.

5. Please report all imaging and image processing related settings as laid out in the small animal micro-CT imaging guideline paper by Bouxsein et al, J Bone Miner Res. 2010 Jul;25(7):1468-86. doi: 10.1002/jbmr.141. This would include x-ray tube energy, tube current, frame averaging settings, image filtration settings, etc.

> We agree and have included the requested details on page 31, lines 771-777.

*6. Please describe how you measured SC BMD. Is that the bone mineral in the TV or in the BV? Also, it is not clear to me why you are using scXRM to define BMC. As you are already measuring BMD with micro-CT, BMC is simply BMD times TV, or BV depending on your normalization. Also, it would be good to show that these numbers are consistent, which means that correlation of BMD*TV or BMD*BV should be highly correlated. Is that the case?*

> Thank you for noting this issue. SC BMC was determined by scXRM as it represents a very simple and rapid technique that includes the majority of the subchondral bone plate. By contrast, ICE μ CT determines BMD within the SC BV in a defined ROI. We agree that these parameters are related and this is confirmed by Spearman correlation analysis: scXRM determined relative scBMC (unit; median grey level) versus ICE μ CT determined scBMD (unit; mg.HA/ccm): lateral tibial plateau, R=0.60; *p<0.001, medial tibial plateau; R=0.541, ***P<0.001.**

7. When reading through your introduction as a representative of the general readership of your manuscript, it is not clear to me how you chose the 50 unselected lines. I find it quite astonishing that you found an OA phenotype in 50% of the lines.

Would you expect to find even more relevant genes in the 1000s of other lines that would still need to be phenotyped? If that is the case, I am not so sure why it is important to discuss potential implications of these genes in such detail but solely based on literature findings without any mechanistic insight. It will necessary to help the reader better understand your rational for the choice of lines.

> Thank you for highlighting this point. As indicated in response 1 to reviewer 1, we clarify that mutant lines were randomly selected. This is because the Sanger Institute, as part of the IMPC, is generating knockout mice for all protein coding genes in an unselected order and we receive samples as the mice are generated through the IMPC pipeline. The question about the frequency of abnormal joint phenotype parameters (>2.0 SD from reference mean) is important. The 50% frequency noted refers to the number of mouse lines that have a single outlier phenotype parameter out of the 18 parameters analysed. In fact this frequency is similar to that identified by rapid-throughput analysis of bone phenotypes using 19 bone structure and strength parameters in a series of 1,000 unselected knockout mice generated by the IMPC (our unpublished data) and is similarly reported in Kemp et al (ref. 17 in the manuscript) and Morris et al (Ref. 19 in the manuscript).

The reason for developing a robust and unbiased prioritisation pipeline to identify mice with outlier joint phenotypes was to ensure that only those with multi-parameter abnormalities, biological evidence, plausibility and relevance to human disease were selected. This pipeline resulted in prioritisation of 4/50 or 8% of randomly selected knockout mice, which are thus discussed in the manuscript.

8. Along those lines, it is not clear to me why you chose to generate a Crispr/Cas9 mutant mice with a Thr92Ala polymorphism in the Dio2 gene. Wouldn't it have been more interesting to generate mutate mice along the pathways of the genes that you have identified in your phenotyping approach to demonstrate proof-of-concept?

> The Dio2 mutant mice were chosen because (i) the mice had already been generated by our collaborators and were available (see ref. 74 in the manuscript)

and (ii) they represent a model of a common human single nucleotide polymorphism that had already been associated with osteoarthritis. Thus, they provided an ideal opportunity to demonstrate the more general applicability of the OBCD joint pipeline in phenotyping mouse models of human disease.

9. One limitation is that you only focused on male mice. No rationale was given. Do your findings hold true for female mice as well? By the way, on page 36, under Skeletal Phenotyping, you say female mice. I guess this is a typo. Otherwise, the text would have to be adapted in the results section.

> Thank you. This point was similarly raised by reviewer 1 in point 4. We have clarified the reasons for focusing on male mice on page 28, lines 688-689 and lines 702-707. The use of females under skeletal phenotyping was not a typographical error – we have clarified this by referring to bone phenotyping and explained the reason why bone phenotyping was performed in females on page 28, lines 702-707.

10. You mention that “Details on backcrossing status, weight, health status, administered drug and procedures, husbandry and specific conditions (including housing, food, temperature and cage conditions) have been reported previously (88,89) ...”. First, I think it is maybe a little much to ask the reader to find this data in these other publications, and second, you should mention whether there are any significant differences between the lines in any these parameters but especially for weight, which might explain differences in the arthritic phenotype as well. Some of these measures may be added to Supplementary Table 1 and discussed in the results section together with the other differences.

> We thank the reviewer for highlighting this point. All of the knockout mice were generated on the same genetic background at the Wellcome Trust Sanger Institute as part of the IMPC pipeline in which standardised husbandry conditions and health status etc. have been comprehensively documented and published previously. As these data apply equally to all the mutants in this study it would not be appropriate to include it again in the current manuscript.

Nevertheless, we agree that increased body weight is an important risk factor for osteoarthritis and we are grateful that the reviewer highlighted our omission. We now include details of body weight for wild-type mice in Supplementary Table 3 and for the 50 knockout lines in Supplementary Table 4. Body weight did not correlate with any phenotype parameter in wild-type mice (Supplementary Table 6). Furthermore body weight was normal in all knockout mice except for *Unk* mice, in which it was decreased (page 8, lines 169-172 and page 13, lines 326-327) and thus unlikely to influence the joint phenotype.

11. You mentioned that this is rapid-throughput imaging approach. Too me, I cannot see a lot of automation in your image analysis approach and therefore would have to conclude that your approach is very labor intensive. You should give numbers of how much time is needed for one sample to be phenotyped for each of the techniques as well as how much time it took from receiving a batch of samples until you had the final results.

> The reviewer highlights an important point. We have noted the 10 steps in the image analysis that include automation throughout the methods section, as requested. We have also now included a comprehensive time and cost analysis of the OBCD rapid-throughput techniques in comparison to the current gold-standard OARSI histology analysis as detailed in response 2 to reviewer 1.

12. On page 29, you mention “data not shown”. I thought that current good practice has changed and that all data mentioned needs to be shown or referenced. Otherwise, the text should be omitted.

> Thank you for noting this; we now include the requested data in Supplementary Table 13.

13. On page 30, you introduce two thresholds in mg/hydroxyapatite/cm³ to either delineate cartilage or bone, respectively. What is the influence of the contrast agent on the choice of threshold?

> The contrast agent (Lipiodol) does not influence the choice of the two thresholds because cartilage cannot be segmented in the absence of contrast medium. Nevertheless, scanning in Lipiodol does have a consistent effect on subchondral bone parameters as detailed below. However, because all scans of wild-type and mutant samples are performed in Lipiodol this does not affect the ability to detect outlier phenotypes since relative parameters are reported as SD differences compared to wild-type (page 32, line 801).

Average (n=2)	BV (mm ³)	BV/TV	Tb.N (1/mm)	Tb.Th (mm)	BMD (mg.HA/ccm)
PBS	0.231	0.574	14.235	0.091	971.195
Lipiodol	0.268	0.686	15.667	0.108	1204.628
Increase (%)	16.19%	19.62%	10.06%	19.31%	24.04%

14. Please label all Tables in the supplementary part. Now there are only the table titles but not the actual table number.

Ralph Müller

> Thank you for noting this omission, which has now been corrected.

Reviewer #3 (Remarks to the Author):

In accordance with the Editor's request, this reviewer provides a review with aspects related to CRISPR. I leave the evaluation of other aspects of the manuscript to other referees.

In the context of CRISPR gene editing, Thr92 and Ala92 mutant mice were used in this study. This reviewer found that these mice have previously been created and validated (ref. #74). Since some of the authors of #74 are identical with that of this

manuscript, not the newly created mice but the previously established ones were supposedly used in this study. Therefore, almost no methodological and scientific description was found in the manuscript concerning CRISPR gene editing. I just provide a few suggestions to improve the manuscript.

Specific comments:

1. If this reviewer's understanding is correct, Thr92 and Ala92 mutant mice used in this study were already reported previously. However, there are many inappropriate description throughout the manuscript. For example, in Abstract, the authors described: "we generate mutant mice with an osteoarthritis-associated polymorphism in the Dio2 gene by Crispr/Cas9 genome editing and demonstrate..." To avoid misunderstanding, the sentence should be corrected, for example - "we used the previously established mutant mice..." Similarly, the whole manuscript should be revised accordingly. If the knock-in mice were newly created in this study, thorough description of the creation and validation of mutant mice must be added.

> Thank you, the reviewer is correct – the mutant mice were reported previously for our studies in reference 74 of this manuscript, and we have amended the manuscript text as requested on page 2, lines 44-46; pages 4-5, lines 105-106; page 30, lines 735-738.

2. In accordance with the description in #74, the wild-type gene of mouse Dio2 contains Pro92, but such explanation was not found in the current manuscript. The authors should add some more background regarding Dio2 gene.

> The reviewer is correct on this point and we are grateful for this being noted. We have included the requested information on page 30, lines 735-738.

3. Both "CRISPR" and "Crispr" were found in the manuscript. The terminology should be unified as "CRISPR."

> We thank the reviewer for pointing out this error and have corrected it throughout the manuscript.

Reviewers' Comments:

Reviewer #1:

Remarks to the Author:

Thank you for a very complete response to the critique and the substantial revisions that have significantly improved the manuscript. One last minor revision is needed to Figure 1. The figure still uses the term "unselected" while the manuscript text has been revised to "randomly selected". The figure should be revised to match the text in the manuscript.

Richard Loeser

Reviewer #2:

Remarks to the Author:

The authors have been able to address almost all my comments and concerns in a satisfactory fashion. Here are some issues that still need addressing though.

1. In Figure 10a, you show that Max Cg.Th is significantly smaller (about 80-84%) for ICEuCT than for histology. Does this lead to significant differences between the two measures, when using a t-test? What is the reason for this difference? In Figure 10a, you used a different notation for Max Cg.Th. I would recommend that you delete the dot behind Max so that you have the same usage throughout the manuscript. Also, in Figure 10c: There is an extra parentheses "(" before the word Histology in both plots' Y-axis that is not required and can be deleted or else needs to be closed after the term ICEuCT.

2. This extra analysis is much appreciated. While almost all of the parameters look very good for precision error and ICC, Cg.DamageArea is less convincing with high PEs and sometimes very low ICC. This can also be seen in Supplementary Figure 10c, where the variance for the reproducibility study is on the same range that the natural variation but individual animals can have much higher values. I am not sure that such a parameter can serve as a reliable quantitative phenotype. I am also not so sure what you have learned from this parameter comparing the different models and the genetic effects of cartilage damage. This might have to be eliminated from the analysis.

6. Thanks for the explanation, which looks like did not find its way into manuscript. I have not seen any changes in the manuscript here. I am still not convinced that I understand what you measure. The BMC from scXRM is clear although maybe quite erroneous. Not everything fast is also good, especially since you are scanning with uCT anyways. These calibrated scans will be much more reliable and accurate. Nevertheless, I am still not happy with your explanations and the way you correlated the values. You cannot easily compare BMC and BMD as one is the mass and the other the density (mass normalized by volume). As mentioned previously you will have to correlate your BMC to $BMD \cdot TV$ if you measured the density in the ROI including BV and bone marrow. If you only determined BMD in the thresholded BV this parameter should not be called BMD but TMD (Tissue Mineral Density) also measured in mg.HA/ccm. In any case, I would expect your correlations to improve. Please improve the description of your measurements to make clearer what you actually measured.

7. I now better understand your approach but I think the limitation remains that you will find many more lines when going through the newly generated mice. I am wondering how it is helpful to discuss those four lines in such detail when they only serve as a successful example of your phenotyping pipeline.

8. The reason for choosing this specific line outside of your target group should be mentioned in the text and for me is a further limitation of the study worthwhile mentioning. It would have been nice to demonstrate that you can take this approach from high-throughput to the analysis of your individual lines. This way this seems quite an arbitrary choice.

13. Again, I think this is important information that you be included in your manuscript. If not only I will know this but the interested reader (whom I am representing) will not have the benefit of your explanation. This will be important for anybody who would like to reconstruct your pipeline and therefore will have to pay attention to these details.

POINT-BY-POINT RESPONSE TO REVIEWERS' COMMENTS

Reviewers' comments:

Reviewer #1 (Remarks to the Author): Richard Loeser

1. Thank you for a very complete response to the critique and the substantial revisions that have significantly improved the manuscript. One last minor revision is needed to Figure 1. The figure still uses the term "unselected" while the manuscript text has been revised to "randomly selected". The figure should be revised to match the text in the manuscript.

> Thank you for spotting our oversight, we have corrected Figure 1 as requested.

Reviewer #2 (Remarks to the Author) (Ralph Müller)

The authors have been able to address almost all my comments and concerns in a satisfactory fashion. Here are some issues that still need addressing though.

1. In Figure 10a, you show that Max Cg.Th is significantly smaller (about 80-84%) for ICEuCT than for histology. Does this lead to significant differences between the two measures, when using a t-test? What is the reason for this difference? In Figure 10a, you used a different notation for Max Cg.Th. I would recommend that you delete the dot behind Max so that you have the same usage throughout the manuscript. Also, in Figure 10c: There is an extra parentheses "((" before the word Histology in both plots' Y-axis that is not required and can be deleted or else needs to be closed after the term ICEuCT.

> We thank the reviewer for noting this important point. As requested we have performed a *t*-test to compare Max Cg.Th determined by ICE μ CT and histology, and the result showed no significant difference for the MTP ($P=0.826$, paired *t*-test, Source Data 12 row 56). Nevertheless, the LTP Max Cg.Th was

significantly thicker when measured by histology (mean value +4.4%) when compared to measurement by ICE μ CT (* $P=0.014$). We have added these data as Supplementary Figure 10e and to Source Data 12.

To investigate this difference, we tested whether ICE μ CT sample preparation resulted in shrinkage. However, ICE μ CT analysis of fresh and fixed samples showed no significant difference in cartilage volume and thickness (Supplementary Figure 10f and Source Data 13). Overall, these studies demonstrate an absence of shrinkage during ICE μ CT sample preparation, but a consistent difference in Max Cg.Th observed in the LTP but not the MTP when comparing ICE μ CT with histology. These findings suggest a difference in anatomical orientation of the LTP relative to the MTP that is revealed when 2D (histology) and 3D (ICE μ CT) imaging are compared. We hypothesise that the apparent increase in LTP thickness determined by histology results from a horizontal anterior-posterior tilt in the LTP relative to the MTP, but this relative anatomical relationship does not affect the Max Cg.Th measured in 3 dimensions by ICE μ CT. We now discuss this point in the Methods section (Lines 869-881).

We have corrected the typographical errors in Supplementary Figure 10 as requested.

2. This extra analysis is much appreciated. While almost all of the parameters look very good for precision error and ICC, Cg. Damage Area is less convincing with high PEs and sometimes very low ICC. This can also be seen in Supplementary Figure 10c, where the variance for the reproducibility study is on the same range that the natural variation but individual animals can have much higher values. I am not sure that such a parameter can serve as a reliable quantitative phenotype. I am also not so sure what you have learned from this parameter comparing the different models and the genetic effects of cartilage damage. This might have to be eliminated from the analysis.

> The ICCs for Cg. Damage Area are 0.818/0.906 (LTP/MTP; WT mice) and 0.952/0.984 (LTP/MTP; mutant mice; Supplementary Table 14). The lowest ICC recorded in this study is 0.818 (LTP Cg. Damage Area, Supplementary Table 14), which is well within the range defined by previous studies as highly reproducible (0.754-0.998; Stok et al., 2016, Reference 83 in the manuscript). Thus, the variation in Cg. Damage Area is due to the very low level of cartilage damage in wild type mice (0-2.8%) as determined by the highly sensitive JSR method that correlates well with OARSI histology. Importantly, this technique provides reliable and sensitive detection of increased cartilage damage, which is a pathognomonic feature of osteoarthritis resulting from DMM provocation or genetic modification (e.g. *Pitx1*^{+/-}). Nevertheless, we agree that the method may be less reliable for quantitation of decreased cartilage damage except in mice with protective mutations that have been challenged by DMM provocation surgery. We now discuss this important limitation (Lines 450-457).

6. *Thanks for the explanation, which looks like did not find its way into manuscript. I have not seen any changes in the manuscript here. I am still not convinced that I understand what you measure. The BMC from scXRM is clear although maybe quite erroneous. Not everything fast is also good, especially since you are scanning with uCT anyways. These calibrated scans will be much more reliable and accurate. Nevertheless, I am still not happy with your explanations and the way you correlated the values. You cannot easily compare BMC and BMD as one is the mass and the other the density (mass normalized by volume). As mentioned previously you will have to correlate your BMC to BMD*TV if you measured the density in the ROI including BV and bone marrow. If you only determined BMD in the thresholded BV this parameter should not be called BMD but TMD (Tissue Mineral Density) also measured in mg.HA/ccm. In any case, I would expect your correlations to improve. Please improve the description of your measurements to make clearer what you actually measured.*

> We did determine bone mineral density in the thresholded BV as noted by the reviewer and have changed the name of this parameter to TMD and

clarified the Methods section (Line 844) as requested. The reviewer is correct that the correlations improve when BMC is compared to BMD*TV, and we have now included and discussed these data in Supplementary Table 6, Source Data 11 (columns AG-AH) and in the Methods (Lines 972-974) as requested.

7. I now better understand your approach but I think the limitation remains that you will find many more lines when going through the newly generated mice. I am wondering how it is helpful to discuss those four lines in such detail when they only serve as a successful example of your phenotyping pipeline.

> Large scale GWAS have demonstrated that many genes are involved in complex-genetic disorders such as osteoarthritis and osteoporosis, which have similar heritability. Indeed, the identification of one or more outlier joint parameters in 50% of knockout lines examined is in accordance with the frequency of outlier parameters identified during bone phenotyping of randomly-selected IMPC knockout mice (our unpublished data, and ref. 16, 17, 19). The four knockout lines highlighted in the manuscript were prioritised by a robust and unbiased pipeline that identified only those with multi-parameter joint abnormalities, biological evidence, plausibility and relevance to human disease (8% of all KO lines). We have included additional text in the Discussion to clarify this important point (Lines 465-473).

8. The reason for choosing this specific line outside of your target group should be mentioned in the text and for me is a further limitation of the study worthwhile mentioning. It would have been nice to demonstrate that you can take this approach from high-throughput to the analysis of your individual

> As requested we now highlight the reason for analysing the *Dio2*^{A1a92} mice in the Results section (Lines 398-399). To demonstrate the broad utility of the joint phenotyping pipeline we describe four distinct applications (i) Screening

of randomly selected KO lines (ii) Analysis of knockout mice with deletion of genes differentially expressed in human OA cartilage, (iii) Analysis of mice with age-related joint degeneration and (iv) Analysis of a mouse model of a common human single nucleotide polymorphism associated with osteoarthritis (Lines 459-463)

13. Again, I think this is important information that you be included in your manuscript. If not only I will know this but the interested reader (whom I am representing) will not have the benefit of your explanation. This will be important for anybody who would like to reconstruct your pipeline and therefore will have to pay attention to these details.

> We apologise for our omission and have now included the information in Supplementary Table 13, Source Data 14 and in the Results section (Lines 845-849).

Reviewers' Comments:

Reviewer #2:

Remarks to the Author:

Thank you for the now very complete response to my comments and the additional revisions that have further improved the manuscript. I have no further requests.